# Impacts of anxiety and socioeconomic factors on mental health in the early phases of the COVID-19 pandemic in the general population in Japan: A web-based survey

Miwako Nagasu[1]*, Kaori Muto[2], Isamu Yamamoto[3]

1 Faculty of Economics, Keio University, Tokyo, Japan, 2 The Institute of Medical Science, The University of Tokyo, Tokyo, Japan, 3 Faculty of Business and Commerce, Keio University, Tokyo, Japan

* mnagasu555@gmail.com

**Data Availability Statement:** All relevant data are within the manuscript and its Supporting Information files.

**Funding:** This work was supported by university grants allocated to the Department of Public Policy,

## Abstract

Owing to the rapid spread of the severe acute respiratory syndrome coronavirus 2 (SARS-CoV-2) pandemic worldwide, individuals experience considerable psychological distress daily. The present study aimed to clarify the prevalence of psychological distress and determine the population most affected by risk factors such as the pandemic, socioeconomic status (SES), and lifestyle-related factors causing psychological distress in the early phases of the pandemic in Japan. This study was conducted via a web-based survey using quota sampling to ensure representativeness of the Japanese population aged 20–64 years. A cross-sectional study of 11,342 participants (5,734 males and 5,608 females) was conducted using a self-administered questionnaire that included the Japanese version of the Kessler 6 Psychological Distress Scale (K6) and questions related to the pandemic, SES, and lifestyle. The prevalence of psychological distress, represented by a K6 score of 5 or more, was 50.3% among males and 52.6% among females. Both males and females with annual household incomes less than 2 million yen and males aged in their twenties had significantly higher K6 scores than those with annual household incomes above 2 million yen and males aged over 30 years. Binary logistic regression analyses found pandemic-related factors such as medical history, inability to undergo clinical tests immediately, having trouble in daily life, unavailability of groceries, new work style, and vague anxiety; SES-related factors such as lesser income; and lifestyle-related factors such as insufficient rest, sleep, and nutritious meals to be significantly related to psychological distress. Psychological distress was more prevalent among people with low income and in younger generations than among other groups. There is an urgent need to provide financial, medical, and social support to those affected by the coronavirus disease 2019 (COVID-19) pandemic.

## Introduction

The coronavirus disease 2019 (COVID-19) pandemic is rapidly spreading worldwide, with a drastic increase in the number of infected patients and related deaths [1, 2]. The mortality rate

Human Genome Centre, Institute of Medical Sciences, University of Tokyo, and by Grand-in-Aid for Scientific Research from the Ministry of Education, Culture, Sports, Science and Technology, Japan (Grant No. 18K01659 and No. 17H06086, Recipient: Isamu Yamamoto). However, the funders had no role in study design, data collection and analysis, decision to publish, or preparation of the manuscript.

**Competing interests:** The authors have declared that no competing interests exist.

due to coronavirus disease 2019 (COVID-19) infections exceeded 1,210,000 people worldwide on 4 November 2020. Since the World Health Organization (WHO) declared the outbreak of COVID-19 disease in January 2020, it has severely and directly affected people's lives and physical health and triggered various psychological problems, such as panic disorder, anxiety, and depression [3–5]. A previous study in China reported that about 35% of people experienced psychological distress and mild-level depression caused the COVID-19 pandemic [3, 6]. Depression was also reported among 27.1% of respondents, and life satisfaction was decreased [7, 8]. Furthermore, the pandemic not only affects physical health conditions but also triggers psychological health such as anxiety and fear of the COVID-19 disease in people worldwide, and preventive measures such as quarantine and new lifestyle practices affect people's mental health [3, 9, 10]. During the outbreak of SARS-CoV-2 in Taiwan in 2003, higher levels of depression were observed among people who, along with their family or friends, were quarantined and were suspected to have SARS-CoV-2 infection [11]. During the early phases of the COVID-19 pandemic in Japan, Muto et al. reported that 85% of Japanese citizens started practicing social distancing measures recommended by the Japanese government [12]. This implies that people have reduced direct communication with other people. These new lifestyles may influence various aspects of mental health conditions among people.

Thus, in this study, we attempted to determine the causes of psychological distress in people during the early phases of the COVID-19 pandemic in Japan. Previous studies have reported that certain psychological problems often occur during similar infectious disease outbreaks [13, 14] or natural disasters [15–17]. Japan has not suffered emerging infectious diseases such as severe acute respiratory syndrome (SARS), Middle East respiratory syndrome (MERS), or the Ebola virus; however, it has survived natural disasters such as the 2011 earthquake off the Pacific coast of Tōhoku [15]. The pandemic has a strong impact on the daily life. In order to take preventive measures, it is essential to identify the risk factors for psychological distress.

Furthermore, there are concerns that the economy may face a recession due to the pandemic. A previous study revealed that mental health worsens and suicidal risks increase during a recession [18]. Historical data suggest that the global financial crisis and natural disasters have increased suicide rates [18]. The COVID-19 pandemic will be a cause of anxiety, depression, increasing alcohol and drug consumption, and suicidal behavior from a public health perspective. Suicide is one of the major causes of mortality, accounting for nearly 1 million deaths per year worldwide [18]. Rana reported that over 300 suicides have already occurred during the lockdown against COVID-19 as "non-coronavirus deaths" due to mental torment in India [19]. In particular, Japan, with 16.6 suicides per 100,000 people, has the fifth highest suicide rate among the Organization for Economic Co-operation and Development (OECD) countries [20]. Mental illnesses and socioeconomic status (SES)-related variables, such as low income and unemployment, were significantly associated with higher suicide risk [21]. Moreover, the results of two studies in the UK and Finland reported a strong association between present financial difficulties and poor mental health conditions [22]. During the COVID-19 pandemic, numerous people received less income or lost their jobs due to the precautionary measures taken. The associations between mental health conditions, the pandemic, SES, and lifestyle-related factors should be identified.

It has been consistently reported that the prevalence of mental illnesses and lifestyle-related factors differs between men and women. In this study, data analysis was stratified by sex [23]. Matud et al. reported that adherence to traditional gender roles had a greater impact on psychological well-being [24]. With regard to lifestyle, smoking is a gender specific risk factor for depression; it has been reported that Japan has more male smokers than female smokers [20]. Gender differences in healthy lifestyles could potentially play confounding roles in the

association between mental health conditions and lifestyle factors, necessitating gender stratification of analyses.

We conducted a web-based survey in the early phases of the COVID-19 pandemic in Japan. In this study, we analysed a large nationally representative data set and hypothesized that the pandemic-, SES-, and lifestyle-related factors would be significantly associated with mental health conditions even after controlling for all potential risk factors. Therefore, the objectives of this study was to identify the factors causing anxiety due to the pandemic-, SES-, and lifestyle-related factors with psychological distress (K6 score $\geq$ 5) among Japanese general population aged 20 to 64 years in the early phases of the pandemic in Japan. Moreover, the associations for both genders were examined separately.

## Materials and methods

### Survey design and participants

As the number of infected people increased gradually, this cross-sectional survey was conducted from 26 to 28 March 2020 via an online platform of a research company (MACRO-MILL INC, Japan). The platform has a pool of approximately 1.2 million registered individuals residing in Japan. We invited a total of 11,342 males and females aged between 20 and 64 years to participate in the survey. During the recruitment process, a quota sampling method was applied. The sample distributions across gender (male or female), age group (20s, 30s, 40s, 50s, or 60s), and employment status (regular employees, non-regular employees, self-employed, or not working) were similar to those of a representative Japanese population. This distribution was based on statistics from the Labour Force Survey (Ministry of Internal Affairs and Communications).

This survey was a closed survey. MACROMILL INC announced an invitation to registered people to participate in this survey via the Internet. Before asking participants to complete the questionnaire, the usability, technical functionality, consistency of the questions, and completeness of the electronic questionnaire were tested by all the authors and MACROMILL INC. All items have been randomized to prevent bias. The E-questionnaire included 51 questions in total. There was a back button to review the answers, and the respondents were able to review or change their answers. This survey automatically eliminated duplicate answers from a single respondent. Incomplete questionnaires were excluded from the analyses. We did not measure how long it took to complete the questionnaire because the respondents had to answer all the questions before proceeding to the next page or completing the questionnaire. At the end of the questionnaire, the participants received compensation.

### Questionnaire and data analysis

**Socioeconomic status (SES)-related factors.** Socioeconomic factors included gender, age, marital status, employment status, educational background, the level of annual disposable income per household, and medical history of the participants. Based on age, the participants were divided into five groups: 20–29 years, 30–39 years, 40–49 years, 50–59 years, and $\geq$ 60 years. Marital status was categorized into two groups: *single* or *married*. On the basis of employment status, four groups were created: *regular employees*, *non-regular employees*, *self-employed and others*, and *not working*. Educational background was classified into three groups: *university or graduate school*, *junior college*, and *high school or junior high school*. All respondents provided information regarding their annual disposable household income in the previous year. This disposable income excluded tax and social insurance fees. The level of disposable income per household was divided into three groups: *less than 2 million yen*, *2 million yen to < 6 million yen*, and *6 million yen and above*. We used the same categorization method

as in the Japanese government survey, the National Survey Health and Nutrition (Ministry of Health, Labour and Welfare, Japan) in 2014. Moreover, we asked the medical history of the participants and noted whether the participant visited the hospital regularly: *No*, *I do not*, and *Yes*, *I go to a hospital regularly*.

**Pandemic-related factors.**   In total, 11 questions related to the pandemic were addressed by the authors considering the situation in Japan: Have you been worried about the following items after the outbreak of the new coronavirus infection? 1. Vague anxiety without a particular reason 2. Anxiety about the possibility that I get infected. 3. Anxiety about the possibility that my family get infected, 4. Inability to receive COVID-19 tests immediately. 5. Lack of medicine. 6. Having trouble in daily life. 7. Unavailability of masks. 8. Lack of groceries, toilet paper, tissue paper, etc. 9. Delays in children's education. 10. Impact on financial conditions such as income. 11. New work styles, such as telework and remote work. The participants answered the 11 questions, with each item scored on a 5-point scale (1 = *extremely worried*, 2 = *slightly worried*, 3 = *Neither or not applicable*, 4 = *I am not much worried*, 5 = *no worries*), and two subscales: Yes (1 and 2 = *Extremely* and *slightly worried*) and No (3, 4, and 5 = *neither*, *not much*, and *no* worries).

**Lifestyle-related factors.**   This study included five lifestyle-related questions to assess daily lifestyle practices: taking sufficient rest and sleep, having nutritious meals, exercising alone such as long distance running and exercises using DVDs, and drinking and smoking habits. For taking a rest and sleep, nutritious meals, and performing exercises, the question was "Which among the following are you practicing to prevent infections? 1) Sufficient rest and sleep. 2) Having nutritious meals. 3) Performing exercises alone, such as long distance running and exercises using DVDs. The responses were categorized into two groups: Yes (1 = *highly applicable* and 2 = *slightly applicable*) and No (3 = *neither*, 4 = *not applicable*, and 5 = *not applicable at all*). Considering the smoking habits, respondents were categorized as *non-smokers*, *ex-smokers*, or *current smokers*. Alcohol consumption was assessed by the question: "How often do you drink alcohol?" The answers were categorized into the following three groups: *never*, ≤ *2 days/week*, and ≥ *3 days/week*.

**Psychological distress assessment.**   The Japanese version of the Kessler Screening Scale for Psychological Distress (K6) was used to screen and assess the severity of mental health problems. The K6 has been shown to have cross-cultural reliability and validity [25]. This scale includes six questions for participants to rate the frequency at which they feel 1) nervous, 2) hopeless, 3) restless or fidgety, 4) so depressed that nothing could cheer them up, 5) that everything was an effort, and 6) worthless during the past 30 days. The scoring system used was as follows: the response categories (1 = *always*, 2 = *almost*, 3 = *sometimes*, 4 = *just a little*, and 5 = *not at all*) were converted into corresponding values (1 = *4 points*, 2 = *3 points*, 3 = *2 points*, 4 = *1 point*, and 5 = *0 points*) to calculate the total score. The total score (ranging from 0 to 24 for K6) has been used to indicate severe mental disorders [26] or mood and anxiety disorders [27]. High scores indicate more severe mental disorders. The participants were then divided into two groups: those with high scores (poor mental health conditions: ≥ 5 points) and those with low scores (good mental health conditions: ≤ 4 points).

## Statistical analysis

The gender-stratified associations of the pandemic, SES, and lifestyle factors with mental health conditions were analysed using Pearson's chi-squared test. The differences in K6 scores in age groups, employment status, and household disposable income groups stratified by sex were analysed using the Kruskal–Wallis test with Bonferroni correction and Student's *t*-test. *P*-values less than 0.05 indicated statistical significance. We investigated the adjusted

prevalence odds ratios (AORs) and 95% confidence intervals (CIs) of high K6 scores ($\geq 5$ points) using binary logistic regression analysis. All SES-related factors (age, marital status, employment status, educational backgrounds, annual disposable income per household, and medical history), pandemic-related factors, and lifestyle-related factors were adjusted. The data were analysed by gender. Statistical analysis was carried out using the SPSS 25 computer package (IBM, Chicago, IL, USA).

### Ethical issues

This survey was conducted in accordance with the Ethical Principles for Sociological Research of the Japan Sociological Society. This approach does not require ethical reviews. There are no national guidelines for social and behavioural research in Japan. This survey does not apply the Japanese government's Ethical Guidelines for Medical and Health Research Involving Human Subjects.

Before starting the web-based questionnaire survey, all participants gave consent to participate in the anonymous online survey by MACROMILL INC. After being informed about the purposes of the survey and their right to quit the survey, all of the participants agreed to participate. They were also provided with the option of "I do not want to respond" for all questions. Completion of the entire questionnaire was considered as indicating the participant's consent. To ensure participants' privacy, we, the authors of this paper, did not obtain any personal information about the participants via the company. MACROMILL INC protects participants' personal information and prevents unauthorised access.

### Results

This study included 11,342 respondents, of whom 5,734 (50.6%) were men and 5,608 (49.4%) were women. The mean age of the males (43.5 ± 12.0 years) was similar to that of the females (43.3 ± 12.0 years). Table 1 shows the sex-specific distribution of the pandemic, SES, lifestyle, and mental health variables. The results indicated statistically significant gender differences in the ratios of the corresponding categories of all variables except age and new workstyles such as telework and remote work. About 56.5% of males and 60.3% of females lived with their partners. Moreover, 69.5% of males and 34.2% of females worked as regular employees. In the context of SES variables, of those with an annual disposable income of less than 2 million yen, 6.7% were male and 8.6% were female. Furthermore, 32.6% of men and 30.3% of women answered that they visited the hospital regularly. Considering the pandemic factors, more females experienced vague anxiety than males (males: 60.1%, females: 74.6%); were concerned about their own health (males: 73.3%, females: 81.3%) and family's health (males: 77.2%, females: 86.2%); lacked immediate access to COVID-19 tests (males: 61.5%, females: 74.5%) and medicine (males: 78.0%, females: 88.6%); have troubles in daily life (males: 72.2%, females: 81.5%) such as unavailability of masks (males: 66.0%, females: 80.8%), some groceries and toilet paper (males: 51.0%, females: 57.4%), delay in children's education (males: 49.5%, females: 51.7%), and impact on economic conditions (males: 72.1%, females: 80.3%); all these differences were statistically significant. Turning to the lifestyle-related variables, more females than males took sufficient rest and sleep, consumed nutritious meals, skipped physical exercises and did not smoke or consume alcohol; these differences were statistically significant. The overall rate of those with a high K6 score was 51.5%. Stratified by sex, 50.3% of the males and 52.6% of the females, respectively, had high GHQ-12 scores (K6: $\geq 5$ points). These gender differences were statistically significant.

Table 2 shows the differences in K6 scores by gender, and in particular by stratified age group, employment status, and annual household income. By gender, the K6 score was

**Table 1. Demographic characteristics of the respondents by gender.**

| | Total | | Male | | Female | | |
|---|---|---|---|---|---|---|---|
| | *n* | % | *n* | % | *n* | % | *P*-value[1] |
| Age (years) | | | | | | | |
| 20–29 | 1964 | 17.3% | 994 | 17.3% | 970 | 17.3% | |
| 30–39 | 2336 | 20.6% | 1196 | 20.9% | 1140 | 20.3% | |
| 40–49 | 3098 | 27.3% | 1568 | 27.3% | 1530 | 27.3% | n.s. |
| 50–59 | 2754 | 24.3% | 1373 | 23.9% | 1381 | 24.6% | |
| ≥ 60 | 1190 | 10.5% | 603 | 10.5% | 587 | 10.5% | |
| Marriage status | | | | | | | |
| Single | 4722 | 41.6% | 2493 | 43.5% | 2229 | 39.7% | *** |
| Married | 6620 | 58.4% | 3241 | 56.5% | 3379 | 60.3% | |
| Employment status | | | | | | | |
| Regular employee | 5906 | 52.1% | 3986 | 69.5% | 1920 | 34.2% | |
| Non-regular employee | 2776 | 24.5% | 733 | 12.8% | 2043 | 36.4% | *** |
| Self-employed and others | 660 | 5.8% | 422 | 7.4% | 238 | 4.2% | |
| Not working | 2000 | 17.6% | 593 | 10.3% | 1407 | 25.1% | |
| Educational background | | | | | | | |
| University or graduate school | 5009 | 44.2% | 3207 | 55.9% | 1802 | 32.1% | |
| Junior college | 2849 | 25.1% | 891 | 15.5% | 1958 | 34.9% | *** |
| High school or junior high school | 3484 | 30.7% | 1636 | 28.5% | 1848 | 33.0% | |
| Household annual income | | | | | | | |
| ≥ 6000K JPY | 3716 | 43.5% | 2218 | 47.4% | 1498 | 38.7% | |
| 2000K– < 6000K JPY | 4186 | 49.0% | 2149 | 45.9% | 2037 | 52.6% | *** |
| < 2000K JPY | 646 | 7.6% | 312 | 6.7% | 334 | 8.6% | |
| Medical history | | | | | | | |
| No, I do not | 7772 | 68.5% | 3863 | 67.4% | 3909 | 69.7% | ** |
| Yes, I go to the hospital regularly | 3570 | 31.5% | 1871 | 32.6% | 1699 | 30.3% | |
| Anxiety after the outbreak | | | | | | | |
| Vague anxiety without a particular reason. | | | | | | | |
| No | 3709 | 32.7% | 2287 | 39.9% | 1422 | 25.4% | *** |
| Yes | 7633 | 67.3% | 3447 | 60.1% | 4186 | 74.6% | |
| Anxiety about the possibility that I get infected. | | | | | | | |
| No | 2578 | 22.7% | 1529 | 26.7% | 1049 | 18.7% | *** |
| Yes | 8764 | 77.3% | 4205 | 73.3% | 4559 | 81.3% | |
| Anxiety about the possibility that my family get infected. | | | | | | | |
| No | 2082 | 18.4% | 1309 | 22.8% | 773 | 13.8% | *** |
| Yes | 9260 | 81.6% | 4425 | 77.2% | 4835 | 86.2% | |
| Inability to receive COVID-19 tests immediately. | | | | | | | |
| No | 3635 | 32.0% | 2207 | 38.5% | 1428 | 25.5% | *** |
| Yes | 7707 | 68.0% | 3527 | 61.5% | 4180 | 74.5% | |
| Lack of medicine | | | | | | | |
| No | 1904 | 16.8% | 1263 | 22.0% | 641 | 11.4% | *** |
| Yes | 9438 | 83.2% | 4471 | 78.0% | 4967 | 88.6% | |
| Having trouble in daily life | | | | | | | |
| No | 2630 | 23.2% | 1593 | 27.8% | 1037 | 18.5% | *** |
| Yes | 8712 | 76.8% | 4141 | 72.2% | 4571 | 81.5% | |
| Unavailability of masks | | | | | | | |
| No | 3027 | 26.7% | 1950 | 34.0% | 1077 | 19.2% | *** |

*(Continued)*

**Table 1.** (Continued)

| | Total | | Male | | Female | | |
|---|---|---|---|---|---|---|---|
| | *n* | % | *n* | % | *n* | % | *P*-value[1] |
| Yes | 8315 | 73.3% | 3784 | 66.0% | 4531 | 80.8% | |
| Lack of groceries, toilet paper, tissue paper, etc. | | | | | | | |
| No | 5200 | 45.8% | 2811 | 49.0% | 2389 | 42.6% | *** |
| Yes | 6142 | 54.2% | 2923 | 51.0% | 3219 | 57.4% | |
| Delay in children's education | | | | | | | |
| No | 5608 | 49.4% | 2897 | 50.5% | 2711 | 48.3% | * |
| Yes | 5734 | 50.6% | 2837 | 49.5% | 2897 | 51.7% | |
| Impact on financial conditions such as income | | | | | | | |
| No | 2706 | 23.9% | 1599 | 27.9% | 1107 | 19.7% | *** |
| Yes | 8636 | 76.1% | 4135 | 72.1% | 4501 | 80.3% | |
| New work styles, such as telework and remote work | | | | | | | |
| No | 8299 | 73.2% | 4167 | 72.7% | 4132 | 73.7% | n.s. |
| Yes | 3043 | 26.8% | 1567 | 27.3% | 1476 | 26.3% | |
| Practice to prevent infection | | | | | | | |
| Sufficient rest and sleep | | | | | | | |
| Yes | 8289 | 73.1% | 3918 | 68.3% | 4371 | 77.9% | *** |
| No | 3053 | 26.9% | 1816 | 31.7% | 1237 | 22.1% | |
| Having nutritious meals | | | | | | | |
| Yes | 7879 | 69.5% | 3680 | 64.2% | 4199 | 74.9% | *** |
| No | 3463 | 30.5% | 2054 | 35.8% | 1409 | 25.1% | |
| Doing exercises that can be done alone | | | | | | | |
| Yes | 3235 | 28.5% | 1796 | 31.3% | 1439 | 25.7% | *** |
| No | 8107 | 71.5% | 3938 | 68.7% | 4169 | 74.3% | |
| Drinking habits | | | | | | | |
| Never | 3980 | 35.1% | 1620 | 28.3% | 2360 | 42.1% | *** |
| $\leq$ 2days/week | 3856 | 34.0% | 2017 | 35.2% | 1839 | 32.8% | |
| $\geq$ 3 days/week | 2646 | 23.3% | 1755 | 30.6% | 891 | 15.9% | |
| Quit | 860 | 7.6% | 342 | 6.0% | 518 | 9.2% | |
| Smoking habits | | | | | | | |
| Never | 6748 | 59.5% | 2628 | 45.8% | 4120 | 73.5% | *** |
| Quit | 2188 | 19.3% | 1415 | 24.7% | 773 | 13.8% | |
| Sometimes + Everyday | 2406 | 21.2% | 1691 | 29.5% | 715 | 12.7% | |
| K6 score | | | | | | | |
| $\leq$ 4 points (Good) | 5506 | 48.5% | 2847 | 49.7% | 2659 | 47.4% | * |
| $\geq$ 5 points (Poor) | 5836 | 51.5% | 2887 | 50.3% | 2949 | 52.6% | |

1) *P*-value from Pearson's chi-squared test

* $P < 0.05$

** $P < 0.01$

*** $P < 0.001$

significantly higher for women than for men. For both sexes, the K6 scores were significantly higher in the 20–29 year old group, and tended to decrease with age. Considering employment status, the K6 score among males who were not working was significantly higher than that of other groups. This tendency was not statistically significant among women. Regarding levels of annual household income, both men and women with a low income of less than 2 million

**Table 2. Differences in K6 scores by gender, age groups, employment status, and household annul income.**

| | Sex | | P value[1] | Age | | | | | | | | | | P value[2] | | | | | Employment status | | | | | | | | P value[3] | | | Household annual income | | | | | | P value[4] | | |
|---|---|---|---|---|---|---|---|---|---|---|---|---|---|---|---|---|---|---|---|---|---|---|---|---|---|---|---|---|---|---|---|---|---|---|---|---|---|---|---|
| | Mean | SD | | 20–29 | | 30–39 | | 40–49 | | 50–59 | | ≥60 | | 30–39 | 40–49 | 50–59 | ≥60 | | Regular employee | | Non regular employee | | Self-employed and others | | Not working | | Non regular employee | Not working | Self-employed and others | <2,000K | | 2,000K–<6,000K | | ≥6,000K | | 2,000K–<6,000K | ≥6,000K | |
| | | | | Mean | SD | Mean | SD | Mean | SD | Mean | SD | Mean | SD | | | | | | Mean | SD | Mean | SD | Mean | SD | Mean | SD | | | | Mean | SD | Mean | SD | Mean | SD | | | |
| Male | 6.23 | 5.975 | | 8.14 | 6.543 | 7.02 | 6.136 | 6.01 | 5.920 | 5.40 | 5.483 | 3.97 | 4.562 | | | | | | | 6.02 | 5.844 | 6.50 | 6.048 | 5.81 | 5.845 | 7.58 | 6.631 | | | | 8.81 | 6.701 | 6.7 | 6.046 | 5.29 | 5.537 | | | |
| | | | | | | | | | | | | | | 20–29: ** | 20–29: *** | 20–29: *** | 20–29: *** | | Regular employee: n.s. | | Non regular employee: n.s. | | Self-employed and others: n.s. | | Not working: *** | | <2,000K: *** | 2,000K–<6,000K: ** | ≥6,000K: *** |
| | | | | | | | | | | | | | | 30–39: | 30–39: *** | 30–39: *** | 30–39: *** | | Non regular employee | | | | | | | | | | | | | | | | | | | |
| | | | ** | | | | | | | | | | | | 40–49: | 40–49: | 40–49: n.s. | 40–49: *** | | Not working | | | | Self-employed and others: *** | | | | | | | | | | | | | | | |
| | | | | | | | | | | | | | | 50–59: | 50–59: | 50–59: | 50–59: *** | | | | | | | | | | | | | | | | | | | | |
| Female | 6.41 | 5.843 | | 7.35 | 6.180 | 6.97 | 6.088 | 6.62 | 5.894 | 5.67 | 5.493 | 4.96 | 4.920 | | 20–29: n.s. | 20–29: n.s. | 20–29: *** | 20–29: *** | | 6.33 | 5.777 | 6.54 | 5.830 | 6.53 | 6.190 | 6.31 | 5.893 | Regular employee: n.s. | Not working: n.s. | Self-employed and others: n.s. | 8.360 | 7.018 | 6.620 | 5.818 | 5.57 | 5.454 | <2,000K: *** | 2,000K–<6,000K: ** | ≥6,000K: *** |
| | | | | | | | | | | | | | | 30–39: | 30–39: n.s. | 30–39: *** | 30–39: *** | | Non regular employee | | | | Self-employed and others: n.s. | | | | | | | 200K–<600K | | | | | | | | |
| | | | | | | | | | | | | | | 40–49: | 40–49: | 40–49: *** | 40–49: *** | | Not working | | | | | | | | | | | | | | | | | | | | |
| | | | | | | | | | | | | | | 50–59: | 50–59: | 50–59: | 50–59: n.s. | | | | | | | | | | | | | | | | | | | | | |

1) The Shapiro-Wilk test was used to assess normality of data distribution for K6 scores. Mann–Whitney U test. **: $P < 0.01$.

2) Comparing K6 scores among five groups. $P$ value from Kruskal-Wallis test with Bonferroni correction: **$P < 0.01$. ***$P < 0.001$.

3) Comparing K6 scores among four groups. $P$ value from Kruskal-Wallis test with Bonferroni correction: *$P < 0.05$, ***: $P < 0.001$

4) Comparing K6 scores among three groups. $P$ value from Kruskal-Wallis test with Bonferroni correction: **: $P < 0.01$, ***: $P < 0.001$

yen had significantly higher K6 scores than those of the other groups. As a result, the high scores were presented by the lowest income groups for both men and women, and males aged 20–29 years and not working, while low scores were presented both by men and women aged 60 years and above, and participants with annual household income of 6 million yen and above.

Table 3 shows the adjusted odds ratios (AOR) from binary logistic regression analyses of all samples and by gender. Among all samples, males and females, for SES factors, employment status such as non-regular employees among females (AOR1.215 [95% CI: 1.025–1.440]) and annual household income (all samples: 2 million–< 6 million: AOR 1.252 [95% CI: 1.133–1.383], < 2 million yen: AOR 1.672 [95% CI: 1.366–2.046], males: 2 million–< 6 million: AOR 1.271 [95% CI: 1.110–1.455], < 2 million yen: AOR 2.036 [95% CI: 1.508–2.749], women: 2 million–< 6 million: AOR 1.201 [95% CI: 1.034–1.396], < 2 million yen: AOR 1.377 [95% CI: 1.034–1.834]) were significantly associated with psychological distress. Having a history of going to a hospital regularly revealed significantly higher AORs (all samples: AOR 1.690 [95% CI: 1.527–1.870], males: AOR 1.720 [95% CI: 1.496–1.978], females: AOR 1.706 [95% CI: 1.468–1.982]). Among all samples and male participants, educational backgrounds such as high school or junior high school (all samples: AOR 1.213 [95% CI: 1.083–1.358] and males: AOR 1.293 [95% CI: 1.114–1.501]) were significantly associated with poor mental conditions.

Considering the pandemic-related factors, feeling vague anxiety without a particular reason (all samples: AOR 1.891 [95% CI: 1.693–2.113], males: AOR 2.026 [95% CI: 1.753–2.341], females: AOR 1.658 [95% CI: 1.390–1.978]), inability to receive COVID-19 tests immediately (all samples: AOR 1.289 [95% CI: 1.144–1.453], males: AOR 1.268 [95% CI: 1.081–1.487], females: AOR 1.309 [95% CI: 1.089–1.574]), having troubles in daily life (all samples: AOR 1.211 [95%CI: 1.067–1.374] and females: AOR 1.414 [95% CI: 1.162–1.721]), lack of groceries, toilet paper, tissue paper, etc. (all samples: AOR 1.258 [95% CI: 1.137–1.391], males: AOR 1.165 [95% CI: 1.012–1.341], females: AOR 1.368 [95% CI: 1.181–1.584]), and new work styles such as telework and remote work (all samples: AOR 1.408 [95% CI: 1.260–1.572], males: AOR 1.421 [95% CI: 1.220–1.654], females: AOR 1.362 [95% CI: 1.157–1.604]) had a significantly greater association with psychological distress than in participants who did not worry about these factors.

Regarding lifestyle-related factors, insufficient rest and sleep (all samples: AOR 1.385 [95% CI: 1.222–1.571], males: AOR 1.459 [95% CI: 1.238–1.720], females: AOR 1.253 [95% CI: 1.028–1.528]), lack of nutritious meals (all samples: AOR 1.243 [95% CI: 1.099–1.405], males: AOR 1.238 [95% CI: 1.052–1.457], females: AOR 1.263 [95% CI: 1.042–1.532]), and female current smokers (AOR 1.227 [95% CI: 1.000–1.505]) were also significantly associated with psychological distress.

Nevertheless, the results indicate the existence of an inverse association between age, marital status, frequent drinking habits, inadequate exercises that can be done alone, such as long distance running and exercises using DVDs, and a factor that may affect the family with psychological distress. The following AOR levels of the age groups were statistically significant (all samples aged 40–49: AOR 0.747 [95% CI: 0.641–0.870], all samples aged 50–59: AOR 0.571 [95% CI: 0.486–0.671], all samples aged 60 or over: AOR 0.365 [95% CI: 0.299–0.445], males aged 40–49: AOR 0.622 [95% CI: 0.503–0.769], males aged 50–59: AOR 0.492 [95% CI: 0.391–0.620], males aged over 60: AOR 0.246 [95% CI: 0.184–0.329], females aged 50–59: AOR 0.664 [95% CI: 0.525–0.840], females aged 60 or over: AOR 0.565 [95% CI: 0.425–0.749]). Being married (all samples: AOR 0.751 [95% CI: 0.676–0.834] and females: AOR 0.637 [95% CI: 0.540–0.753]), frequent drinking habits (all samples: AOR 0.840 [95% CI: 0.741–0.952] and males: AOR 0.832 [95% CI: 0.703–0.986]), lack of exercises that can be performed alone, such as long distance running and exercises using DVDs (all samples: AOR 0.788 [95% CI: 0.710–

**Table 3. Associations of the K6 score with risk factors among all samples and by gender.**

| | All samples | | | Males | | | Females | | |
|---|---|---|---|---|---|---|---|---|---|
| | | 95% CI | | | 95% CI | | | 95% CI | |
| | Adjusted OR | Lower | Upper | Adjusted OR | Lower | Upper | Adjusted OR | Lower | Upper |
| **Sex** | | | | | | | | | |
| Male | 1 | | | | | | | | |
| Female | 0.922 | 0.827 | 1.027 | | | | | | |
| **Age (years)** | | | | | | | | | |
| 20–29 | 1 | | | 1 | | | 1 | | |
| 30–39 | 0.925 | 0.789 | 1.083 | 0.865 | 0.696 | 1.073 | 0.977 | 0.771 | 1.237 |
| 40–49 | **0.747** | **0.641** | **0.870** | **0.622** | **0.503** | **0.769** | 0.91 | 0.726 | 1.141 |
| 50–59 | **0.571** | **0.486** | **0.671** | **0.492** | **0.391** | **0.620** | **0.664** | **0.525** | **0.840** |
| ≥ 60 | **0.365** | **0.299** | **0.445** | **0.246** | **0.184** | **0.329** | **0.565** | **0.425** | **0.749** |
| **Marriage status** | | | | | | | | | |
| Single | **1** | | | 1 | | | 1 | | |
| Married | **0.751** | **0.676** | **0.834** | **0.869** | **0.750** | **1.006** | **0.637** | **0.540** | **0.753** |
| **Employment status** | | | | | | | | | |
| Regular employee | 1 | | | 1 | | | 1 | | |
| Non-regular employee | 1.108 | 0.976 | 1.258 | 0.973 | 0.783 | 1.211 | **1.215** | **1.025** | **1.440** |
| Self-employed and others | 1.023 | 0.836 | 1.253 | 1.075 | 0.833 | 1.388 | 0.966 | 0.680 | 1.370 |
| Not working | 1.128 | 0.979 | 1.300 | 1.191 | 0.931 | 1.525 | 1.136 | 0.938 | 1.377 |
| **Educational background** | | | | | | | | | |
| Univ. or grad. school | 1 | | | 1 | | | 1 | | |
| Junior college | 1.084 | 0.962 | 1.223 | 1.055 | 0.881 | 1.264 | 1.045 | 0.881 | 1.239 |
| High school or junior high school | **1.213** | **1.083** | **1.358** | **1.293** | **1.114** | **1.501** | 1.09 | 0.912 | 1.303 |
| **Household annual income** | | | | | | | | | |
| ≥6,000K JPY | 1 | | | 1 | | | 1 | | |
| 2,000K-<6,000K JPY | **1.252** | **1.133** | **1.383** | **1.271** | **1.110** | **1.455** | **1.201** | **1.034** | **1.396** |
| <2,000K JPY | **1.672** | **1.366** | **2.046** | **2.036** | **1.508** | **2.749** | **1.377** | **1.034** | **1.834** |
| **Medical history** | | | | | | | | | |
| No, I don't. | 1 | | | 1 | | | 1 | | |
| Yes, I go to the hospital regularly. | **1.690** | **1.527** | **1.870** | **1.720** | **1.496** | **1.978** | **1.706** | **1.468** | **1.982** |
| **Anxiety after the outbreak** | | | | | | | | | |
| Vague anxiety without a particular reason. | | | | | | | | | |
| No | 1 | | | 1 | | | 1 | | |
| Yes | **1.891** | **1.693** | **2.113** | **2.026** | **1.753** | **2.341** | **1.658** | **1.390** | **1.978** |
| Anxiety about the possibility that I get infected. | | | | | | | | | |
| No | 1 | | | 1 | | | 1 | | |
| Yes | 1.116 | 0.959 | 1.299 | 1.081 | 0.886 | 1.319 | 1.136 | 0.894 | 1.444 |
| Anxiety about the possibility that my family get infected. | | | | | | | | | |
| No | 1 | | | 1 | | | 1 | | |
| Yes | 0.864 | 0.736 | 1.013 | **0.737** | **0.601** | **0.902** | 1.206 | 0.923 | 1.574 |
| Inability to receive COVID-19 tests immediately. | | | | | | | | | |
| No | 1 | | | 1 | | | 1 | | |

(*Continued*)

**Table 3.** (Continued)

| | All samples | | | Males | | | Females | | |
|---|---|---|---|---|---|---|---|---|---|
| | | 95% CI | | | 95% CI | | | 95% CI | |
| | Adjusted OR | Lower | Upper | Adjusted OR | Lower | Upper | Adjusted OR | Lower | Upper |
| Yes | *1.289* | *1.144* | *1.453* | **1.268** | **1.081** | **1.487** | **1.309** | **1.089** | **1.574** |
| Lack of medicine. | | | | | | | | | |
| No | 1 | | | 1 | | | 1 | | |
| Yes | *0.879* | *0.752* | *1.027* | 0.926 | 0.762 | 1.126 | 0.903 | 0.692 | 1.180 |
| Having trouble in daily life. | | | | | | | | | |
| No | 1 | | | 1 | | | 1 | | |
| Yes | *1.211* | *1.067* | *1.374* | 1.121 | 0.948 | 1.325 | **1.414** | **1.162** | **1.721** |
| Unavailability of masks. | | | | | | | | | |
| No | 1 | | | 1 | | | 1 | | |
| Yes | *0.947* | *0.838* | *1.069* | 0.894 | 0.761 | 1.050 | 1.049 | 0.867 | 1.269 |
| Lack of groceries, toilet paper, tissue paper, etc. | | | | | | | | | |
| No | 1 | | | 1 | | | 1 | | |
| Yes | *1.258* | *1.137* | *1.391* | **1.165** | **1.012** | **1.341** | **1.368** | **1.181** | **1.584** |
| Delay in children's education | | | | | | | | | |
| No | 1 | | | 1 | | | 1 | | |
| Yes | *1.077* | *0.974* | *1.191* | 1.083 | 0.942 | 1.245 | 1.053 | 0.909 | 1.220 |
| Impact on financial conditions such as income | | | | | | | | | |
| No | 1 | | | 1 | | | 1 | | |
| Yes | *1.084* | *0.961* | *1.222* | 1.058 | 0.902 | 1.242 | 1.152 | 0.955 | 1.389 |
| New work styles such as telework and remote work. | | | | | | | | | |
| No | 1 | | | 1 | | | 1 | | |
| Yes | *1.408* | *1.260* | *1.572* | **1.421** | **1.220** | **1.654** | **1.362** | **1.157** | **1.604** |
| Practice to prevent infection | | | | | | | | | |
| Taking rest and sleep sufficiently. | | | | | | | | | |
| Yes | 1 | | | 1 | | | 1 | | |
| No | *1.385* | *1.222* | *1.571* | **1.459** | **1.238** | **1.720** | **1.253** | **1.028** | **1.528** |
| Having nutritious meals | | | | | | | | | |
| Yes | 1 | | | 1 | | | 1 | | |
| No | *1.243* | *1.099* | *1.405* | **1.238** | **1.052** | **1.457** | **1.263** | **1.042** | **1.532** |
| Doing exercises that can be done alone. | | | | | | | | | |
| Yes | 1 | | | 1 | | | 1 | | |
| No | *0.788* | *0.710* | *0.874* | **0.692** | **0.600** | **0.797** | 0.944 | 0.807 | 1.105 |
| Drinking habits | | | | | | | | | |
| Never | 1 | | | 1 | | | 1 | | |
| ≤ 2days/week | *1.014* | *0.907* | *1.134* | 1.007 | 0.857 | 1.182 | 1.031 | 0.880 | 1.208 |
| ≥3 days/week | *0.840* | *0.741* | *0.952* | **0.832** | **0.703** | **0.986** | 0.895 | 0.734 | 1.091 |
| Quit | *1.113* | *0.925* | *1.340* | 1.258 | 0.947 | 1.672 | 1.052 | 0.821 | 1.348 |
| Smoking habits | | | | | | | | | |
| Never | 1 | | | 1 | | | 1 | | |
| Quit | *1.027* | *0.908* | *1.162* | 1.057 | 0.898 | 1.244 | 1.004 | 0.825 | 1.222 |
| Sometimes +Everyday | *1.060* | *0.940* | *1.194* | 0.996 | 0.856 | 1.159 | **1.227** | **1.000** | **1.505** |

0.874] and males: AOR 0.692 [95% CI: 0.600–0.797]), and the factors affecting family (males: AOR 0.737 [95% CI: 0.601–0.902]) showed a significant association with psychological distress.

## Discussion

The results of this study revealed that over half of the participants felt psychological distress (K6: ≥ 5 points) in the early phases of the COVID-19 pandemic in Japan. By gender, the K6 scores among females were significantly higher than among males. In both sexes, the younger groups had significantly higher K6 scores than the older groups. The results revealed that the levels of psychological distress tended to decline with age. Regarding SES factors such as employment status and the levels of annual household income, men who are not working, and both men and women with low incomes (less than 2 million yen) had significantly higher K6 scores than the other groups.

Moreover, according to the results of binary logistic regression analyses after controlling for all covariates, the majority of potential risk factors among males and females were similar. Regarding SES factors, employment status such as regular employees among females and lower level of annual household income among both men and women were significantly associated with psychological distress. A history of regular hospital visits was also a significant risk factor for psychological distress. Regarding the pandemic-related factors, feelings of vague anxiety without a particular reason; inability to receive COVID-19 tests immediately; having troubles in daily life among females; lack of groceries, toilet paper, and tissue paper; and new work styles such as telework and remote work had significantly stronger associations with psychological distress than did participants who did not worry about these factors. Regarding lifestyle-related factors, inadequate rest, sleep, nutritious meals, and current smokers among females were also significantly associated with psychological distress.

Nevertheless, these results indicate the existence of an inverse association between age, marital status, frequent drinking habits, inadequate exercise, and factors that may affect family and psychological distress.

### The prevalence of psychological distress

The prevalence of K6 scores of 5 or more points in this study was higher than in the results of the Comprehensive Survey of Living Conditions in Japan in 2010 and 2007. The Comprehensive Survey of Living Conditions has been conducted by the Ministry of Health, Labour and Welfare for the general population in Japan in a sample of 750,000 since 1986. Comparing the results of the K6 scores among these survey results, the present study demonstrated that 51.5% of all participants (50.3% of men and 52.6% of women) had a K6 score of 5 points or higher. The results of the Comprehensive Survey of Living Conditions in Japan indicated that 28.7% of all participants in 2010 (18.2% among males and 31.1% among females) and 28.9% of all participants in 2007 (26.1% among males and 31.6% among females) demonstrated a K6 score of 5 points or more [28]. These results suggest that more than 20% of people were more affected by psychological distress in the early phases of the pandemic than during the non-pandemic period. Wang et al. also reported that 53.8% of Chinese respondents showed a psychological impact of the outbreak and 28.8% reported anxiety symptoms [29]. The COVID-19 pandemic has had a significant negative impact on mental health.

Regarding gender differences in the K6 scores, the female scores were significantly higher than those of males in this study. Three previous studies in China and Turkey also reported that female respondents reported significantly higher psychological distress than male respondents [3, 6, 30]. A previous study of the impact of the Great East Japan Earthquake in 2011

reported that females were more prone to psychological distress than men [15]. The results of this study are similar to those of these previous studies.

Comparing stress levels among age groups by gender, the present study found that the level of psychological distress tended to decline with age. Younger males and females had significantly higher K6 scores than the older groups. Previous studies reported that psychological distress levels tended to decline with age [30, 31] and found higher stress levels among the young-adult group (18–30 years) [3, 30]. The depression scores of Turkish female participants between 18 and 29 years of age were found to be the highest among age groups [6]. Another study reported that young people in China tend to obtain a large amount of information from social media, which can easily trigger stress [3]. The WHO recommends watching less news about COVID-19, as it may lead to anxiety [1]. Thus, frequent access to social media and news programs among the younger generation might be a trigger for poor mental conditions. More research is needed on the younger generation to clarify the impact of the frequency of checking social media on psychological distress during the pandemic.

Regarding SES factors such as employment status and the levels of annual household income, men who are not working and both men and women with low income had significantly higher K6 scores than the other groups. Moreover, among all the samples and male participants, an educational background of high school or junior high school was significantly associated with poor mental conditions. People who are not working and people with low levels of education tend to have lower income in Japan. Lei et al. reported that not working and lower average household income are significantly associated with higher scores on the self-rating depression scale (SDS) during the COVID-19 pandemic in China [30]. Nagasu et al. reported that low household disposable income was significantly associated with mental health conditions during a non-pandemic situation in Japan [32]. Low income would be a potential risk factor for mental health even during a non-pandemic period; therefore, prompt financial support would be needed by people with lower income to cope with the increasing financial difficulties and anxieties.

Moreover, according to the results of binary logistic regression analyses after controlling for all covariates, the majority of potential risk factors among males and females were similar. A history of regular hospital visits was also a significant risk factor for psychological distress. These results imply that the cause was the dissemination of information that the mortality rate of people who go to a hospital regularly for chronic disease was high. In the early stages of the pandemic, news programmes broadcasted information that people with chronic illness showed a high mortality rate. However, it was not mentioned what kind of chronic disease showed a high mortality rate. The Centers for Diseases Control and Prevention stated that chronic diseases are defined broadly as conditions that last 1 year or more and require ongoing medical attention, limit activities of daily living, or both [33]. Chronic diseases such as heart disease, cancer, and diabetes are the leading causes of death and disability, as well as obesity, hypertension, Alzheimer's disease, Epilepsy, and blindness. It was found that people with diabetes, high blood pressure, and kidney disease would show more severe symptoms when contracting COVID-19 [34]. Examining the types of chronic diseases that lead to a high mortality rate due to COVID-19 is essential.

Considering the pandemic-related factors, feeling vague anxiety without a particular reason; Inability to receive COVID-19 tests immediately; lack of groceries, toilet paper, tissue paper, etc.; and new working styles, such as telework and remote work had a significantly greater association with psychological distress. Until now, Japan has been less affected by infectious diseases such as SARS, MARS, and Ebola haemorrhagic fever; hence, the Japanese people might be generally less resilient against an outbreak of an unknown infectious disease. Particularly during the early phase of the pandemic, when there was no accurate information about

the novel coronavirus, it is likely that feelings of vague anxiety without a particular reason affected psychological health severely. Before the pandemic in Japan, clinical tests were usually available in hospitals. Moreover, groceries and other essential household items were also available; however, due to the pandemic, access to the COVID-19 tests have been limited by the Japanese Government, and groceries are not easily available at supermarkets. Li et al. reported that inadequate supplies of hand sanitizers increased anxiety and depression [5]. The psychological effects of these sudden changes can be observed in people due to anxiety and psychological distress. Furthermore, according to the results, the new working style is considered to have caused a psychological burden. While preparing for the possible second pandemic in the future, it is necessary to prepare for an appropriate clinical test system and stock up groceries [5].

Having troubles in daily life among all groups and female participants and employment status such as regular employees among female participants showed a significant association with psychological distress. It is considered that women, who carry out most of the household chores, including shopping and daily life activities in Japan, particularly those who are regularly employed, were greatly affected by the fact that they could not purchase groceries and their working styles had changed due to the pandemic. Both men and women may need social support for the changes in their lives during the pandemic.

Even in the early stages of the pandemic, inadequate rest, sleep, and nutritious meals were also significantly associated with psychological distress. Regarding female participants, current smokers showed a significant association with poor mental health conditions. A previous study reported that participants who experienced sleep problems or started to smoke and drink alcohol showed moderate levels of depression [6]. Taking adequate rest, sleep, and nutritious meals is essential to boost immunity during the pandemic in the absence of effective drugs and vaccines. In general, previous studies have reported linear relationships between mental health and lifestyle-related factors such as short sleep duration [35], unbalanced diets [36], lack of habitual physical exercise [36, 37], smoking habits [38–40], and alcohol consumption [41]. Even during the pandemic, it is important to establish a regular life with healthy lifestyle practices. The WHO recommends avoiding unhelpful coping strategies such as the use of tobacco and alcohol [1]. Healthy lifestyle practices such as sufficient rest and sleep, eating healthy foods, performing physical exercise, and staying in touch with family and friends would be helpful in reducing anxiety and psychological distress.

Nevertheless, the results revealed inverse associations between age, marital status, frequent drinking habits, lack of physical exercise, and psychological distress. In this study, being married was inversely associated with poor mental health conditions. A Chinese study also reported that married people had lower levels of psychological distress than single people [31]. Being able to talk to a partner during the pandemic may reduce psychological distress, though divorce due to being quarantined at home has become a social problem. Frequent drinking habits, with alcohol consumption 3 days or more per week, showed an inverse association with psychological distress; however, this inverse association between alcohol intake and mental health is controversial. One study also reported that habitual drinking was inversely associated with poor mental health conditions among Japanese adults during the non-pandemic period, and that a nonlinear relationship elevates risks for depression and anxiety in heavy drinkers compared to light and moderate drinkers [42]. Byles et al. reported that moderate alcohol intake may carry some health benefits for older women in terms of survival and quality of life [43]. Appropriate grouping would be required based on the amount of alcohol present in alcoholic beverages, rather than considering drinking frequency. Further research is warranted on the amount of alcohol intake, while problematic drinking would be harmful.

Lack of physical exercise indicated an inverse association with psychological distress. Some previous studies reported that physical activity was a protective factor against developing

depression [44]. Aerobic exercise and moderate-intensity training were most beneficial for psychological well-being in older adults without clinical disorders [45]. During the time of this survey, a cluster of patients presented with a new coronavirus infection at training gyms, which was the main topic in the TV news in Japan every day. Presumably, those who did not exercise had better psychological effects. If the pandemic is prolonged, it will be necessary to consider safe exercise practices.

This study was conducted in the early phase of the COVID-19 pandemic in Japan, and thus makes some important contributions. First, the strength of the study is its use of timely data from a large number of respondents. This was scientifically important as it identified the prevalence of psychological distress and the associations of the pandemic-, SES-, and lifestyle-related factors with mental health outcomes among Japanese samples of over 10,000 respondents. Second, this study found significant associations between the pandemic-, SES-, and lifestyle-related factors and psychological distress after controlling for relevant factors. The results imply that the level of psychological distress was higher in the early phase of the pandemic than in the non-pandemic period. Both male and female participants reported similar and diverse risk factors for psychological distress.

This study has several limitations. First, this survey was a web-based survey. The participants were required to access the questionnaire on the website using a device which can connect the Internet. Members of the general population who did not use the Internet could not participate in the study, thereby causing information bias. Second, the sample was collected using a quota sampling method from individuals who were recruited by or who were self-enrolled in the Internet panel of the online research, and not via random sampling of the whole population of Japan. The quota sampling method ensured a similar distribution to the Japanese population among demographic groups (gender, age, and employment status), but the sample within each group does not necessarily reflect the population. Third, the study design is cross-sectional and thus cannot capture changes or causal relationships between psychological distress and its risk factors over the course of the COVID-19.

## Conclusions

This study aimed to identify the prevalence and associated risk factors of psychological distress among the general population in Japan in the early phases of the COVID-19 pandemic. The findings reveal that the prevalence of psychological distress among people with low incomes and the younger generations is higher than in other groups. There is a need to pay more attention to public psychological distress, especially among young people with low income levels. The results showed that younger generations with low income are more likely to experience anxiety and psychological distress. Various psychological interventions could be organised for the psychological characteristics of different target population groups. As a risk factor for psychological distress, it was found that there is a significant relationship with psychological distress in those who visit the hospital regularly. This is because people with chronic illness and those who visited the hospital regularly were more likely to have fatal symptoms due to the new COVID-19 infection. To relieve psychological distress, it is necessary to examine and provide accurate information on the types of chronic diseases with high fatality rates. Regarding the factors related to the pandemic, it was found that "people are not able to receive the COVID-19 tests immediately" indicated a significant association with psychological distress. At the early stages of the pandemic, the COVID-19 testing system was fragile. Patients were unable to receive COVID-19 tests immediately, even when physicians determined that it was necessary. The result of this study may reflect the poor testing system during the study period —March 2020. However, the system has improved day by day; the numbers of COVID-19

tests taken increased significantly from 13,026 cases in March 15 to 3,035,324 cases in November 15, 2020 [46, 47]. Moreover, a variety of tests are now available. Moreover, vague anxiety without a particular reason showed a significant association with psychological distress, whereas information about unknown SARS-CoV-2 was not reliable and changed often. Furthermore, lifestyle-related factors also increased psychological distress. The results indicate that people felt anxious about new infections due to changes in the daily lives. The impacts of COVID-19 on mental health conditions are potential risk factors. Healthy lifestyle practices need to be established even during the pandemic.

There is an urgent need to provide financial, medical, and social support to those affected by the COVID-19 pandemic. In the early stages of the spread of new infectious diseases, it is essential to prioritise care of people with chronic disease. Providing accurate reasons and information about COVID-19 testing to people who cannot undergo COVID-19 testing may help reducing the level of anxiety. These results suggest that it is necessary to improve the medical system, including COVID-19 testing, and the supply chains of groceries and especially toilet paper, and prepare for future pandemics to reduce anxiety and psychological distress.

## Supporting information

**S1 Questionnaire.**
(DOCX)

**S1 Data.**
(XLSX)

## Acknowledgments

We would like to thank the participants in our online survey for their valuable data. This work was supported by university grants allocated to the Department of Public Policy, Human Genome Centre, Institute of Medical Sciences, University of Tokyo, and by Grand-in-Aid for Scientific Research from the Ministry of Education, Culture, Sports, Science and Technology (No. 18K01659 and No. 17H06086).

## Author Contributions

**Conceptualization:** Miwako Nagasu, Kaori Muto, Isamu Yamamoto.

**Data curation:** Miwako Nagasu, Kaori Muto, Isamu Yamamoto.

**Formal analysis:** Miwako Nagasu.

**Funding acquisition:** Kaori Muto.

**Investigation:** Isamu Yamamoto.

**Project administration:** Kaori Muto, Isamu Yamamoto.

**Supervision:** Kaori Muto, Isamu Yamamoto.

**Writing – original draft:** Miwako Nagasu, Kaori Muto, Isamu Yamamoto.

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
