## [Decision Letter · Decision Letter 0]

27 Jul 2020

PONE-D-20-21244

Impacts of anxiety and socioeconomic factors on mental health in the early phases of the COVID-19 pandemic in the general population in Japan: a Web-based survey

PLOS ONE

Dear Dr. Nagasu,

Thank you for submitting your manuscript to PLOS ONE. After careful consideration, we feel that it has merit but does not fully meet PLOS ONE’s publication criteria as it currently stands. Therefore, we invite you to submit a revised version of the manuscript that addresses the points raised during the review process.

The two reviewers addressed several major and minor concerns about your manuscript. Please revise your manuscript carefully.

We look forward to receiving your revised manuscript.

Kind regards,

Kenji Hashimoto, PhD

Academic Editor

PLOS ONE

Journal Requirements:

https://bmcpublichealth.biomedcentral.com/articles/10.1186/s12889-019-8022-4

In your revision ensure you cite all your sources (including your own works), and quote or rephrase any duplicated text outside the methods section. Further consideration is dependent on these concerns being addressed.

Reviewers' comments:

Reviewer's Responses to Questions

**Comments to the Author**

1. Is the manuscript technically sound, and do the data support the conclusions?

Reviewer #1: Partly

Reviewer #2: No

2. Has the statistical analysis been performed appropriately and rigorously? 

Reviewer #1: N/A

Reviewer #2: No

3. Have the authors made all data underlying the findings in their manuscript fully available?

Reviewer #1: Yes

Reviewer #2: Yes

4. Is the manuscript presented in an intelligible fashion and written in standard English?

Reviewer #1: Yes

Reviewer #2: No

5. Review Comments to the Author

Reviewer #1: The authors aimed to clarify the prevalence of psychological distress and determine the most affected population by risk factors such as the pandemic, socioeconomic status (SES) and lifestyle-related factors with psychological distress in the early phases of the pandemic in Japan.

The strength of the study is its use of timely data from a large number of respondents.

(Prevalence of psychological distress and the association of the pandemic-, SES-, and lifestyle-related factors with mental health outcomes among Japanese samples of over 10,000 respondents.)

This study is interesting, however I have a few suggestions.

#1: The authors asked the medical history of the participants and noted whether the participant visits the hospital regularly.

Could you tell me the department in the field of medical care?

(This may affect the results.)

#2: Would you tell me more data of the method of sampling?

The authors should follow the Checklist for Reporting Results of Internet E-Surveys (CHERRIES) to minimize the potential bias.

Reviewer #2: The author attempted to reveal the influence of socio-economic status on psychiatric burden with COVID-19. The purpose of this study is acceptable. However, there are several flaws in this paper both in the methodology and interpretation of the results.

First of all, it is doubtful that the results of this study reflect the influence of COVID-19. As the author mentions, SES, gender, some lifestyles have been proved to be associated with psychiatric distress. Then, the differences between participants with high distress (K6>=5) and others can be seen before COVID-19 pandemic. Younger people are anxious than elders before the pandemic due to their financial discrepancy, aren't they? Why does the author believe the outcomes of this study were brought from the current pandemic? Are there similar studies before pandemic to compare their data with the present study's quantitatively?

In the introductory section, the author suggests the possibility that COVID-19 would raise suicide rate in Japan. At present, this estimation has not been realized: suicide rate in the 2020 spring in Japan is significantly low compared to those in the past years.

As well, the influence of changed lifestyles is controversial. Some clinicians comment avoiding to go to the office is advantageous for workers who feel stressful in the interpersonal relationship. There may be complicated issues around this phenomenon.

In the method section, there are also many queries.

The author is encouraged to disclose the whole questionnaire sheet as a supplementary file.

The method of implementation of the survey should be disclosed in detail. Were the participants rewarded? Was duplicated answering effectively excluded? How did the author exclude the answers by non-serious respondents?

The author should adhere to the guidelines of web-based survey such as CHERRIES. Relevant information should be disclosed.

Why did the author exclude people over 65 y.o. from the study? Aging is one of the biggest issues in Japanese society. I hardly understand why the author omitted elder people's opinions.

How did the author calculate the sample size? Are ten thousands enough to prove the author's hypothesis?

How did the author set the threshold of disposable income without tax as 2m and 6m JPY?

The author took a series of questionnaire regarding the pandemic-related factors. Some of the items are positively associated with the psychological distress with statistical analysis. However, the results should be cautiously interpreted. To begin with, each item describes a pattern of anxiety. Thus, high score of the questionnaire can be conceptually equal to high score of K6.

Sub-scaling seems arbitrarily developed by the author. It may be not appropriate that the answer of "not applicable" is deemed as not worried. Relationship between items can be complicated. (i.e. participants with no kids never worry about their children's education. But they were likely to unmarried, thus can be more anxious.)

Considering several issues mentioned above, I do not recommend the author to include these items into independent valuables in the binary logistic regression analysis.

Particularly, the question "inability to receive a PCR test immediately" seems problematic. Does the author believe PCR test should be provided to everyone who want to do immediately? There are arguments even among specialists in this theme. PCR can neither provide 100% sensitivity nor specificity. If massive people take the test, there will be many false negative persons, leading to make them super-spreaders. In my sense, many of the participants who answered this question "yes" lack adequate knowledge of medical examination.

In the discussion section, there are some mentioning which are hardly accepted generally.

The author wrote women might be more susceptible to stress than men and would tend to develop mental illness. Are there some evidences to support this statement? Indeed, some kinds of mental illness such as depression is more common in women. But suicide rate is much higher in men. A common interpretation is that women are likely to express their distress to others as well as call for help. If so, men should be cared with more intensity. On the other hand, women with children can be more anxious for their children's health and education. In this sense, the author's conclusion that mental health interventions and treatments for women is needed is not acceptable completely.

The hypothesis that young people watching SNS and news programs become anxious is not supported by this study. The author did not ask the frequency of access to SNS. Considering that this study was web-based, elder participants can also be accustomed to social network. In addition, people over 65 y.o. did not participate in this study.

The sentences "These results imply that the cause was the dissemination of information that the mortality rate of people who go to a hospital regularly for curing chronic disease was high. It is essential to examine the type of chronic diseases that led to a high mortality rate due to COVID-19." does not make sense. Did the author asked the worry about fatality when infected by COVID-19? Simply, people with chronic diseases are likely to be depressed, rich evidence suggests.

The sentences "Up until now, Japan has been less affected by infectious diseases; hence, Japanese people have higher anxiety against the COVID-19." is also hardly understandable. What did the author compare Japanese people with?

Is it true that "groceries are not easily available" in the pandemic situation? Are Japanese suffering from starvation?

The sentences "It is considered that women who carry most of the household chores including shopping and daily life activities in Japan, particularly those who are regularly employed, were greatly affected by the fact that they could not purchase groceries and their working styles changed due to the pandemic." is meaningless. Were single men with regular employment not affected? Did the author confirmed that most married female participant were responsible for daily purchasing?

It is no doubt that healthy lifestyles are beneficial for better mental health. The description in the line 355 - 366 is merely repeating it, not deducting novel findings from the result of this study.

The sentence "Presumably, the person who did not exercise had better psychological effects." is no more than imaginary idea by the author, I have to say, because the author asked the participants about neither frequency of the media exposure nor utilizing a gym.

In the conclusion section, I cannot agree with some descriptions for the reasons mentioned above.

I do not believe "providing accurate information the type of chronic diseases with high fatality rate in people" will relieve people with chronic diseases from distress. (In my personal sense, guaranteeing adequate medical care as well as not pandemic regardless of the social situation is the most important for them, because many of them were required to refrain from, or reluctant to, visiting hospital, for the fear of infection, or simply rack of medical resources.)

As mentioned above, I think "allowing people to undertake the PCR test" is not appropriate definitely, whereas establishment of proper inspection strategy is needed.

Above all, this study protocol cannot be well developed. Also, the author's interpretation of the results are partially biased. Generally speaking, I cannot admit this study as qualified.

Nonetheless, the data relevant with COVID-19 pandemic can be valuable, considering the current confusing situation. I strongly recommend the author to reconsider the whole manuscript, with disclosing methods in detail, cutting biased interpretations, to make it usable for researchers in the future.

6. PLOS authors have the option to publish the peer review history of their article (what does this mean?). If published, this will include your full peer review and any attached files.

Reviewer #1: No

Reviewer #2: **Yes: **Akihiro Shiina

---

## [Author Response · Author response to Decision Letter 0]

9 Nov 2020

Reviewers' comments:

1. #1: The authors asked the medical history of the participants and noted whether the participant visits the hospital regularly. Could you tell me the department in the field of medical care?

(This may affect the results.)

Response:

Thank you for this question. Unfortunately, we did not ask which department the participants regularly visit in this study. 

2. #2: Would you tell me more data of the method of sampling?

The authors should follow the Checklist for Reporting Results of Internet E-Surveys (CHERRIES) to minimize the potential bias.

Response:　Thank you for your recommendation. We followed the CHERRIES and added detailed information to the Methods section as much as possible. Please refer to the Methods section. 

3. Reviewer #2: The author attempted to reveal the influence of socio-economic status on psychiatric burden with COVID-19. The purpose of this study is acceptable. However, there are several flaws in this paper both in the methodology and interpretation of the results. First of all, it is doubtful that the results of this study reflect the influence of COVID-19. As the author mentions, SES, gender, some lifestyles have been proved to be associated with psychiatric distress. Then, the differences between participants with high distress (K6>=5) and others can be seen before COVID-19 pandemic. 1) Younger people are anxious than elders before the pandemic due to their financial discrepancy, aren't they? 2) Why does the author believe the outcomes of this study were brought from the current pandemic? 3) Are there similar studies before pandemic to compare their data with the present study's quantitatively? 

Response:

1) Thank you for your comments. Yes, younger people were. According to our published results, younger people showed more anxiety than elders before the pandemic, too. However, the reason was not only financial discrepancy; there were various other potential risk factors. About this study, the results for the younger generations revealed that the pandemic increased the prevalence of K6 scores of 5 or more. 

2) Because we asked the question: “Do you feel anxiety about the following items after the COVID-19 pandemic?” The answer should be directly related to the impact of the pandemic. The purpose of this paper is to determine how the COVID-19 pandemic-related anxiety factors affected psychological conditions as measured by K6 scores after controlling for the basic characteristics and SES-related factors. 

We rewrote the objective of this study as follows: 

“Therefore, the objectives of this study was to identify the factors causing anxiety due to the pandemic-, SES-, and lifestyle-related factors with psychological distress (K6 score ≥ 5) among Japanese general population aged 20 to 64 years in the early phases of the pandemic in Japan.”

3) We compared the results of this study and the Comprehensive Survey of Living Conditions in Japan in 2010 and 2007 and summarized this issue in the Discussion as follows:

“The prevalence of K6 scores of 5 or more points in this study was higher than in the results of the Comprehensive Survey of Living Conditions in Japan in 2010 and 2007. The Comprehensive Survey of Living Conditions has been conducted by the Ministry of Health, Labour and Welfare for the general population in Japan in a sample of 750,000 since 1986. Comparing the results of the K6 scores among these survey results, the present study demonstrated that 51.5% of all participants (50.3% of men and 52.6% of women) had a K6 score of 5 points or higher. The results of the Comprehensive Survey of Living Conditions in Japan indicated that 28.7% of all participants in 2010 (18.2% among males and 31.1% among females) and 28.9% of all participants in 2007 (26.1% among males and 31.6% among females) demonstrated a K6 score of 5 points or more (28). These results suggest that more than 20% of people were more affected by psychological distress in the early phases of the pandemic than during the non-pandemic period. Wang et al. also reported that 53.8% of Chinese respondents showed a psychological impact of the outbreak and 28.8% reported anxiety symptoms (29). The COVID-19 pandemic has had a significant negative impact on mental health.” 

Reference: 

1. Nagasu M, Kogi K, Yamamoto I. Association of socioeconomic and lifestyle-related risk factors with mental health conditions: a cross-sectional study. BMC Public Health. 2019;19(1):1759.

4. 1) In the introductory section, the author suggests the possibility that COVID-19 would raise suicide rate in Japan. At present, this estimation has not been realized: suicide rate in the 2020 spring in Japan is significantly low compared to those in the past years.

2) As well, the influence of changed lifestyles is controversial. Some clinicians comment avoiding to go to the office is advantageous for workers who feel stressful in the interpersonal relationship. There may be complicated issues around this phenomenon.

Response:

1) Thank you for your comment. Although the suicide rate from February to June 2020 was significantly lower than in the past five years, the suicide rate in July, August, and September increased dramatically. The number of people who committed suicide nationwide in August was 1,849, an increase of 15.3% from the same period last year. We do wish that COVID-19 would not raise the suicide rate in Japan. 

We rewrote the sentence as follows:

“The COVID-19 pandemic will be a cause of anxiety, depression, increasing alcohol and drug consumption, and suicidal behavior from a public health perspective.”

2) Thank you for your comment. Some clinicians told me that telework can be stressful due to lack of communication with colleagues. I also feel that working from home is more stressful than working at an office. As you mentioned, the influence of changed lifestyles and working conditions has been very controversial. That is why we must conduct this study and reveal the impact of the pandemic. 

5. In the method section, there are also many queries.

The author is encouraged to disclose the whole questionnaire sheet as a supplementary file.

Response:

Thank you for your comment. We have already stated what questions we used and how to categorize them in the Method section. As you recommended, we translated the questions used and attached the questionnaire as a supplementary file. 

6. The method of implementation of the survey should be disclosed in detail. Were the participants rewarded? 

Response: 

In accordance with Comment 2, we added detailed information to the Methods section. Yes, the participants gained incentives. We added the sentence as follows:

“At the end of the questionnaire, the participants received compensation.” 

7. Was duplicated answering effectively excluded? 

Response:

Yes, it was. This survey automatically eliminated duplicate answers from a single respondent. We added detailed information on this matter to the Methods section. 

8. How did the author exclude the answers by non-serious respondents?

Response:

This Web survey system did not accept strange answers and advance to the next question. All respondents had to choose from the options displayed and answer all questions properly. Otherwise, if all questions were not answered, they could not go to next question.

9. The author should adhere to the guidelines of web-based survey such as CHERRIES. Relevant information should be disclosed.

Response: 

Thank you for your suggestions. This comment is the same as Comment 2; please refer to our response there.

10. Why did the author exclude people over 65 y.o. from the study? Aging is one of the biggest issues in Japanese society. I hardly understand why the author omitted elder people's opinions.

Response:

Thank you for this question. Because people over the age of 65 usually receive a pension. There are obvious differences in economic conditions under and over 65 years old in Japan. In this study, we were interested in socio-economic factors such as annual disposable household income. Therefore, we decided that our target population should be people not on a pension, and thus our target population was under 65. 

11. How did the author calculate the sample size? Are ten thousands enough to prove the author's hypothesis?

Response: 

It is well known that calculating the sample size from the sample error is suitable for surveys with a limited number of questions. However, when asking a variety of questions and performing a variety of analyses, the required sample size differs depending on the analysis. Therefore, we secured the maximum number of samples allowed by the budget so as to be able to handle cases with a low appearance rate in order to perform more complete analyses.

12. How did the author set the threshold of disposable income without tax as 2m and 6m JPY?

Response:

Thank you for your question. The categorization method is the same as in the Japanese government survey, the National Survey Health and Nutrition (Ministry of Health, Labour and Welfare, Japan) in 2014.

Thus, we added the following sentence:

“We used the same categorization method as in the Japanese government survey, the National Survey Health and Nutrition (Ministry of Health, Labour and Welfare, Japan) in 2014.”

13. The author took a series of questionnaires regarding the pandemic-related factors. Some of the items were positively associated with psychological distress with statistical analysis. However, the results should be cautiously interpreted. To begin with, each item describes a pattern of anxiety. Thus, high score of the questionnaire can be conceptually equal to high score of K6.

Sub-scaling seems arbitrarily developed by the author. It may be not appropriate that the answer of "not applicable" is deemed as not worried. Relationship between items can be complicated. (i.e. participants with no kids never worry about their children's education. But they were likely to unmarried, thus can be more anxious.)

Considering several issues mentioned above, I do not recommend the author to include these items into independent valuables in the binary logistic regression analysis.

Response:

Thank you for the comment. 

1) As you mentioned that it may be inappropriate to interpret the answer “not applicable” as not worried. We also considered your point. It is a common practice to divide the five choices into two categories by looking at the distribution. In this study, we would like to mention that as “neither or not applicable” is an option, “not applicable” cannot be separated from “neither (I do not know)” by the questionnaire design, so we made it “not worried”.

As you mentioned that a relationship between items can be complicated, we believe that there is a need to adjust various variables by using the binary logistic regression analysis. 

We also believe that the participants with children or without children would be concerned about children’s education. We asked, “Delays in children’s education,” not “Delays in your own children’s education,” so it did not ask whether the participants had their own children or not. In order to prevent the spread of the new coronavirus, the Japanese government requested that all schools, elementary school, junior high and high schools, and special needs schools throughout the country be temporarily closed from March 2 until spring break on very short notice. This request affected things such as caregivers’ work shifts at their workplaces. This is why we asked whether the participants worried about children’s education in Japan.

14. Particularly, the question "inability to receive a PCR test immediately" seems problematic. 1) Does the author believe PCR test should be provided to everyone who want to do immediately? There are arguments even among specialists in this theme. PCR can neither provide 100% sensitivity nor specificity. If massive people take the test, there will be many false negative persons, leading to make them super-spreaders. In my sense, many of the participants who answered this question "yes" lack adequate knowledge of medical examination.

Response:

1) No, we don’t. What we would like to do is to provide evidence of how Japanese people feel about the PCR test. The results should form one of the basic materials for policy discussion. For example, if Japanese people feel anxiety when they cannot receive the PCR test immediately, the Japanese Government may need to consider how to solve this problem. One possible solution is to share adequate knowledge of medical examinations, as the reviewer mentioned. 

15. In the discussion section, there are some mentioning which are hardly accepted generally. The author wrote women might be more susceptible to stress than men and would tend to develop mental illness. Are there some evidences to support this statement? Indeed, some kinds of mental illness such as depression is more common in women. But suicide rate is much higher in men. A common interpretation is that women are likely to express their distress to others as well as call for help. If so, men should be cared with more intensity. On the other hand, women with children can be more anxious for their children's health and education. In this sense, the author's conclusion that mental health interventions and treatments for women is needed is not acceptable completely.

Response:

Thank you for your suggestions. We deleted the part mentioned:

“Women might be more susceptible to stress than men and would tend to develop mental illness. There is a need to provide appropriate mental health interventions and treatments for women.”

16. The hypothesis that young people watching SNS and news programs become anxious is not supported by this study. The author did not ask the frequency of access to SNS. Considering that this study was web-based, elder participants can also be accustomed to social network. In addition, people over 65 y.o. did not participate in this study.

Response:

Unfortunately, we did not include a variable about the frequency of checking social media in this study. The Discussion section includes the passage, “Young people in China tend to obtain a large amount of information from social media that can easily trigger stress (3). WHO recommends to watch less news about COVID-19 that may lead to anxieties (1). Frequent access to social media and news programs among young generation would be one of the major causes of poor mental conditions; hence, more research is required for the young generation to clarify what influences psychological distress level during the pandemic.” This was one of interpretations of the results. 

However, we rewrote this part as follows: 

“Another study reported that young people in China tend to obtain a large amount of information from social media, which can easily trigger stress (3). The WHO recommends watching less news about COVID-19, as it may lead to anxiety (1). Thus, frequent access to social media and news programs among the younger generation might be a trigger for poor mental conditions. More research is needed on the younger generation to clarify the impact of the frequency of checking social media on psychological distress during the pandemic.”

We also deleted our Objective No. 1. Our objective in this study as to identify the to the pandemic-, SES-, and lifestyle-related factors associated with psychological distress (K6 score ≥ 5) in the early phases of the pandemic in Japan. 

17. The sentences "These results imply that the cause was the dissemination of information that the mortality rate of people who go to a hospital regularly for curing chronic disease was high. It is essential to examine the type of chronic diseases that led to a high mortality rate due to COVID-19." does not make sense. 1) Did the author asked the worry about fatality when infected by COVID-19? Simply, people with chronic diseases are likely to be depressed, rich evidence suggests.

Response:

1) Thank you for your comment. We have added the following:

“In the early stages of the pandemic, news programmes broadcasted information that people with chronic illness showed a high mortality rate. However, it was not mentioned what kind of chronic disease showed a high mortality rate. The Centers for Diseases Control and Prevention stated that chronic diseases are defined broadly as conditions that last 1 year or more and require ongoing medical attention, limit activities of daily living, or both (34). Chronic diseases such as heart disease, cancer, and diabetes are the leading causes of death and disability, as well as obesity, hypertension, Alzheimer’s disease, Epilepsy, and blindness. We have found that people with diabetes, high blood pressure, and kidney disease would show more severe symptoms when contracting COVID-19 (35). It is essential to examine the types of chronic diseases that lead to a high mortality rate due to COVID-19.”

18. The sentences "Up until now, Japan has been less affected by infectious diseases; hence, Japanese people have higher anxiety against the COVID-19." is also hardly understandable. What did the author compare Japanese people with?

Response: 

Thank you for your comment. We have changed the sentence as follows: 

“Until now, Japan has been less affected by infectious diseases such as SARS, MARS, and Ebola hemorrhagic fever; hence, Japanese people might be generally less resilient against an outbreak of unknown infectious disease.”

19. Is it true that "groceries are not easily available" in the pandemic situation? Are Japanese suffering from starvation?

Response: 

Yes, it was. It is well known that foods such as noodles, pasta, and frozen foods sold out at many supermarkets in March because people bought these foods in order to stock up during the pandemic. The TV news reported that these foods were not avaible in many supermarkets in Tokyo.

Following this line of thought, the Japanese Government has enforced the Act on Emergency Measures for Stabilizing Living Conditions of the Public (the provisions of Articles 26 and 37). The Act was applied to groceries and personal protective equipment such as masks and hand sanitizer on 10 March 2020 and 22 May 2020, respectively. This Act prohibited purchasing large quantities of masks and hand sanitizers for resale as a business. 

According to the results of this study, we are unable to say whether Japanese people were suffering starvation, but it is not an aim of this study. We focused on the association between the availability of groceries and mental health conditions in the early stages of the pandemic. 

20. The sentences "It is considered that women who carry most of the household chores including shopping and daily life activities in Japan, particularly those who are regularly employed, were greatly affected by the fact that they could not purchase groceries and their working styles changed due to the pandemic." is meaningless. 1) Were single men with regular employment not affected? 2) Did the author confirmed that most married female participant were responsible for daily purchasing?

Response:

1) Thank you for your comment. We stated the results regarding Table 3 before the sentence you cited above: “Having troubles in daily life among all groups and female participants and employment status such as regular employees among female participants showed a significant association with psychological distress.” The psychological distress of male respondents showed no significant associations with single status, regular employment, or troubles in daily life. The results imply that these variables were not associated with psychological distress among men. Thus, we cannot say that male respondents with single status or regular employment suffered psychological distress.

2) We do not think that we need to confirm that most married female participant were responsible for daily purchasing. Based on our results, both male and female respondents indicated associations between psychological distress and “lack of groceries, toilet paper, tissue paper, etc.” Thus, we wrote that “Both men and women may need social support for the changes in their lives during the pandemic.”

21. It is no doubt that healthy lifestyles are beneficial for better mental health. The description in the line 355 - 366 is merely repeating it, not deducting novel findings from the result of this study.

Response:

Thank you for your suggestion. One of novel findings of this study is that healthy lifestyles conduce to better mental health during the pandemic. There are many publications about lifestyle in non-pandemic situations, but publications on the early stages of a pandemic are very limited. Therefore, we believe that these results are valuable. Thus, we have added the following sentence:

“Even in the early stages of the pandemic, inadequate rest, sleep, and nutritious meals were also significantly associated with psychological distress.” 

22. The sentence "Presumably, the person who did not exercise had better psychological effects." is no more than imaginary idea by the author, I have to say, because the author asked the participants about neither frequency of the media exposure nor utilizing a gym.

Response:

Thank you for your comment. Unfortunately, we did not include variables about the frequency of media exposure or gym attendance in this study. However, please note the results in Table 3 for the variable “Doing exercises that can be done alone,” which indicate that “the person who did not exercise alone had better psychological effects.” As we wrote in the discussion part, “During the time of this survey, a cluster of patients presented with a new coronavirus infection at training gyms, which was the main topic in the TV news in Japan every day.” Thus, the results of this study indicate that many people might have stopped exercising because of a cluster of patients in gyms. In addition, at that time it was unclear how to prevent the infection in gyms. Thus, our results show that those who did not exercise showed better psychological effects. 

23. In the conclusion section, I cannot agree with some descriptions for the reasons mentioned above.

I do not believe "providing accurate information the type of chronic diseases with high fatality rate in people" will relieve people with chronic diseases from distress. (In my personal sense, guaranteeing adequate medical care as well as not pandemic regardless of the social situation is the most important for them, because many of them were required to refrain from, or reluctant to, visiting hospital, for the fear of infection, or simply rack of medical resources.)

Response:

1) Thank you for your comment. I apologize for repeating the same explanation as for Comment 17. 

According to our results, in the early stages of the pandemic, news programmes repeatedly broadcasted that people with chronic illness showed a high mortality rate. As a result, a majority of people refrained from going to the hospital, and even now people still hesitate to go to the hospital, because we had not identified what kinds of chronic disease can increase the risk of death. Nowadays, the Japanese Government has stated that people suffering from diabetes, high blood pressure, and kidney disease will experience more severe symptoms when infected with COVID-19. Thus, we wrote, “To relieve psychological distress, it is necessary to examine and provide accurate information the type of chronic diseases with high fatality rate in people.” 

24. As mentioned above, I think "allowing people to undertake the PCR test" is not appropriate definitely, whereas establishment of proper inspection strategy is needed.

Response:

Thank you for your idea. We also agree that the “establishment of proper inspection strategy is needed.” However, according to our results and what we wrote in the Conclusion, “it was found that ‘people are not able to undergo polymerase chain reaction (PCR) test by themselves according to their will’ indicated a significant association with psychological distress. It is essential to urgently establish an inspection system that allows people to undertake the PCR test.” We very much understand the current situation in Japan. There are various reasons and difficulties why all people cannot receive the PCR test, but people demand receiving the PCR test as a human right as a Japanese citizen. We believe that nobody can deny a human right, although there are some difficulties in providing opportunities to take the PCR test. Therefore, the Japanese Government needs to establish proper inspections as soon as possible. 

25. Above all, this study protocol cannot be well developed. Also, the author's interpretation of the results are partially biased. Generally speaking, I cannot admit this study as qualified.

Nonetheless, the data relevant with COVID-19 pandemic can be valuable, considering the current confusing situation. I strongly recommend the author to reconsider the whole manuscript, with disclosing methods in detail, cutting biased interpretations, to make it usable for researchers in the future.

Response: 

Thank you for the comment. We have faithfully read your comments and replied.

---

## [Decision Letter · Decision Letter 1]

25 Nov 2020

PONE-D-20-21244R1

Impacts of anxiety and socioeconomic factors on mental health in the early phases of the COVID-19 pandemic in the general population in Japan: A web-based survey

PLOS ONE

Dear Dr. Nagasu,

Thank you for submitting your manuscript to PLOS ONE. After careful consideration, we feel that it has merit but does not fully meet PLOS ONE’s publication criteria as it currently stands. Therefore, we invite you to submit a revised version of the manuscript that addresses the points raised during the review process.

The reviewer #2 did not satisfy the response to the comments. Please revise your manuscript again. We may need another reviewer to make the final decision for your revised manuscript.

We look forward to receiving your revised manuscript.

Kind regards,

Kenji Hashimoto, PhD

Academic Editor

PLOS ONE

Reviewers' comments:

Reviewer's Responses to Questions

**Comments to the Author**

1. If the authors have adequately addressed your comments raised in a previous round of review and you feel that this manuscript is now acceptable for publication, you may indicate that here to bypass the “Comments to the Author” section, enter your conflict of interest statement in the “Confidential to Editor” section, and submit your "Accept" recommendation.

Reviewer #1: All comments have been addressed

Reviewer #2: (No Response)

2. Is the manuscript technically sound, and do the data support the conclusions?

Reviewer #1: Yes

Reviewer #2: No

3. Has the statistical analysis been performed appropriately and rigorously? 

Reviewer #1: I Don't Know

Reviewer #2: Yes

4. Have the authors made all data underlying the findings in their manuscript fully available?

Reviewer #1: Yes

Reviewer #2: Yes

5. Is the manuscript presented in an intelligible fashion and written in standard English?

Reviewer #1: Yes

Reviewer #2: Yes

6. Review Comments to the Author

Reviewer #1: Thank you for your polite reply.

1. #1: The authors asked the medical history of the participants and noted whether the participant visits the hospital regularly. Could you tell me the department in the field of medical care?

(This may affect the results.)

Response:

Thank you for this question. Unfortunately, we did not ask which department the participants regularly visit in this study.

＝＞ Thank you for your polite answer.

2. #2: Would you tell me more data of the method of sampling?

The authors should follow the Checklist for Reporting Results of Internet E-Surveys (CHERRIES) to minimize the potential bias.

Response: Thank you for your recommendation. We followed the CHERRIES and added detailed information to the Methods section as much as possible. Please refer to the Methods section.

=> Thank you for your polite reply.

Reviewer #2: The author has properly amended the description of the original manuscript in almost all areas with argument.

However, I never understand why the author claims that "people demand receiving the PCR test as a human right as a Japanese citizen."

To begin with, the author themselves wrote they don't believe PCR test should be provided to everyone who want to do immediately in the former section of the response letter. Their comments are quite paradoxical.

I am doubtful that the author understand the sensibility and specificity of an examination. The current sensitivity of PCR is estimated as 70%. Regarding specificity, there is controversy, estimated between 99% to 99.99%. Even if the specificity is 99.999%, ten people of a million without COVID-19 will receive false positive, leading to unnecessary seclusion and wasting medical resource. False negative case can be more serious. Discussing providing PCR alone is quite risky and unrealistic.

Again, I have to say that the outcome of the present survey does not support the idea that allowing any people to undertake the PCR test is appropriate. It is an issue of public health, not human right. Human right is an essential matter which should never been violated. It is far from protecting human right to justify things the majority wants to do. If the vast majority of Japanese support racism, should the Japanese government take such a policy? Ridiculous. As far as the author use the term "human right" in an arbitrary manner, the author's manuscript does not deserve to be read.

7. PLOS authors have the option to publish the peer review history of their article (what does this mean?). If published, this will include your full peer review and any attached files.

Reviewer #1: No

Reviewer #2: **Yes: **Akihiro Shiina

---

## [Author Response · Author response to Decision Letter 1]

24 Dec 2020

Authors’ response to reviewers:

PLOS ONE

PONE-D-20-21244

Impacts of anxiety and socioeconomic factors on mental health in the early phases of the COVID-19 pandemic in the general population in Japan: a Web-based survey

Thank you very much for reviewing our manuscript. We have made the suggested changes as follows:

6. Review Comments to the Author

Reviewer #1: Thank you for your polite reply.

1. #1: The authors asked the medical history of the participants and noted whether the participant visits the hospital regularly. Could you tell me the department in the field of medical care?

(This may affect the results.)

Response:

Thank you for this question. Unfortunately, we did not ask which department the participants regularly visit in this study.

＝＞ Thank you for your polite answer.

Response:

Thank you for your time to review our manuscript. 

2. #2: Would you tell me more data of the method of sampling?

The authors should follow the Checklist for Reporting Results of Internet E-Surveys (CHERRIES) to minimize the potential bias.

Response: Thank you for your recommendation. We followed the CHERRIES and added detailed information to the Methods section as much as possible. Please refer to the Methods section.

=> Thank you for your polite reply.

Response:

Thank you for your comment. 

Reviewer #2: The author has properly amended the description of the original manuscript in almost all areas with argument.

However, I never understand why the author claims that "people demand receiving the PCR test as a human right as a Japanese citizen."

To begin with, the author themselves wrote they don't believe PCR test should be provided to everyone who want to do immediately in the former section of the response letter. Their comments are quite paradoxical.

I am doubtful that the author understand the sensibility and specificity of an examination. The current sensitivity of PCR is estimated as 70%. Regarding specificity, there is controversy, estimated between 99% to 99.99%. Even if the specificity is 99.999%, ten people of a million without COVID-19 will receive false positive, leading to unnecessary seclusion and wasting medical resource. False negative case can be more serious. Discussing providing PCR alone is quite risky and unrealistic.

Again, I have to say that the outcome of the present survey does not support the idea that allowing any people to undertake the PCR test is appropriate. It is an issue of public health, not human right. Human right is an essential matter which should never been violated. It is far from protecting human right to justify things the majority wants to do. If the vast majority of Japanese support racism, should the Japanese government take such a policy? Ridiculous. As far as the author use the term "human right" in an arbitrary manner, the author's manuscript does not deserve to be read.

Response:

Thank you for your time to review our manuscript. 

Please accept our sincere apology regarding improper terminology usage and taking up your crucial time.

First, we sincerely apologize for using the word “human rights” in an unthoughtful manner. If you allow us, we would like to remove the word and the corresponding sentence, mentioned in our previous response, Comment 24, to reviewers.

Second, we deleted the word “PCR tests.” Our original question was “F_15: Have you been worried about the following items after the outbreak of the new coronavirus infection?” The sub-question we provided to the respondents was “3) Inability to receive COVID-19 tests immediately.” In the early stages of the pandemic in Japan, “COVID-19 tests” meant “PCR tests”; however, writing “PCR tests” is not accurate, and is different from the original question. Thus, we deleted the word “PCR tests” and rewrote as “COVID-19 tests” in the manuscript. In addition, we also rewrote “at will” to “immediately,” in an attempt to describe the questions more accurately.

Third, we realized that people’s circumstances at the time this survey was conducted (March) were different from when this manuscript was written (November). Therefore, we added more information on the differences between the testing systems in place during March and November 2020 as follows: 

“At the early stages of the pandemic, the COVID-19 testing system was fragile. Patients were unable to receive COVID-19 tests immediately, even when physicians determined that it was necessary. The result of this study may reflect the poor testing system during the study period—March 2020. However, the system has improved day by day; the numbers of COVID-19 tests taken increased significantly from 13,026 cases in March 15 to 3,035,324 cases in November 15, 2020 (47, 48). Moreover, a variety of tests are now available.”

“Providing accurate reasons and information about COVID-19 testing to people who cannot undergo COVID-19 testing may help reducing the level of anxiety.”

---

## [Decision Letter · Decision Letter 2]

11 Jan 2021

PONE-D-20-21244R2

Impacts of anxiety and socioeconomic factors on mental health in the early phases of the COVID-19 pandemic in the general population in Japan: A web-based survey

PLOS ONE

Dear Dr. Nagasu,

Thank you for submitting your manuscript to PLOS ONE. After careful consideration, we feel that it has merit but does not fully meet PLOS ONE’s publication criteria as it currently stands. Therefore, we invite you to submit a revised version of the manuscript that addresses the points raised during the review process.

Additional reviewer #3 addressed several minor concerns about your manuscript. Pleaser revise your manuscript again.

We look forward to receiving your revised manuscript.

Kind regards,

Kenji Hashimoto, PhD

Academic Editor

PLOS ONE

Reviewers' comments:

Reviewer's Responses to Questions

**Comments to the Author**

1. If the authors have adequately addressed your comments raised in a previous round of review and you feel that this manuscript is now acceptable for publication, you may indicate that here to bypass the “Comments to the Author” section, enter your conflict of interest statement in the “Confidential to Editor” section, and submit your "Accept" recommendation.

Reviewer #3: (No Response)

2. Is the manuscript technically sound, and do the data support the conclusions?

Reviewer #3: Yes

3. Has the statistical analysis been performed appropriately and rigorously? 

Reviewer #3: (No Response)

4. Have the authors made all data underlying the findings in their manuscript fully available?

Reviewer #3: No

5. Is the manuscript presented in an intelligible fashion and written in standard English?

Reviewer #3: Yes

6. Review Comments to the Author

Reviewer #3: Thank you for giving me a chance to review this manuscript. I confirmed the authors have already amended the questions from the other reviewers in the previous rounds. The paper now is well written and has scientific meaning. Therefore, I suggest its acceptance after fixing several points below.

Major:

1. I can understand that the authors changed the expressions of “PCR tests” to “COVID-19 tests” to answer previous reviewer 2’s question. However, I think it is not acceptable to change the questionnaire after it has been answered. Because if we ask a question in another way, the response may be different. Authors may add some explanation about “PCR tests here were the only tests for COVID-19 at that time.” However, I do not think it is acceptable to change the original question.

2. Supplemental data for all the response data were not included in this revision, while PLOS ONE asked authors to upload their data somewhere. I found the data in revision R1. I did not know why authors deleted them.

Minor points:

3. P.17 L.229 Regular employees among females (AOR 1.125 …)

While in Table 3 (P.20), it is “Non regular” employee (AOR “1.215”). Please confirm both the label and the number, and fix either one.

4. P.17 L.230-232 there are typos. Shouldn't all “≥ 6 million” be “< 6 million?” (according to Table 3.)

5. P.20 Table 3 Some names of rows were difficult to be understood and different from the questionnaire, e.g., “What I may be infected with the virus,” “What may affect my family.”

6. P.22 L.326 “Three previous studies in China and Turkey also reported that Chinese female …”

The study from Turkey (ref. 6) should not report Chinese results. Therefore, please remove “Chinese” here.

7. P. 22-23 “Previous research concluded that females were at a higher risk of mental health problems than men (31).”

Ref. 31 is a paper on post-injury mental health problems. It seems to be a particular case and a little far from the current results. Could the authors revise this part? Also, in the following sentence, “these three previous studies” is written. It is better to remove “three,” if the authors mention both Ref 15 and 31, together with Ref 3, 6, and 30.

8. P. 28 L.430 “using a PC device” via the internet.

To my knowledge, such kind of survey can also be answered by smartphones. If so, please remove using a PC device or add some other means to make it clear.

7. PLOS authors have the option to publish the peer review history of their article (what does this mean?). If published, this will include your full peer review and any attached files.

Reviewer #3: **Yes: **Yu-Shi Tian

---

## [Author Response · Author response to Decision Letter 2]

7 Feb 2021

Response to reviewers:

PLOS ONE

PONE-D-20-21244

Impacts of anxiety and socioeconomic factors on mental health in the early phases of the COVID-19 pandemic in the general population in Japan: a Web-based survey

Reviewers' comments:

Reviewer's Responses to Questions

Comments to the Author

1. If the authors have adequately addressed your comments raised in a previous round of review and you feel that this manuscript is now acceptable for publication, you may indicate that here to bypass the “Comments to the Author” section, enter your conflict of interest statement in the “Confidential to Editor” section, and submit your "Accept" recommendation.

Reviewer #3: (No Response)

2. Is the manuscript technically sound, and do the data support the conclusions?

Reviewer #3: Yes

3. Has the statistical analysis been performed appropriately and rigorously? 

Reviewer #3: (No Response)

4. Have the authors made all data underlying the findings in their manuscript fully available?

Reviewer #3: No

5. Is the manuscript presented in an intelligible fashion and written in standard English?

Reviewer #3: Yes

6. Review Comments to the Author

Reviewer #3: Thank you for giving me a chance to review this manuscript. I confirmed the authors have already amended the questions from the other reviewers in the previous rounds. The paper now is well written and has scientific meaning. Therefore, I suggest its acceptance after fixing several points below.

Response:

Thank you very much for reviewing our manuscript. We have made the suggested changes.

Major:

1. I can understand that the authors changed the expressions of “PCR tests” to “COVID-19 tests” to answer previous reviewer 2’s question. However, I think it is not acceptable to change the questionnaire after it has been answered. Because if we ask a question in another way, the response may be different. Authors may add some explanation about “PCR tests here were the only tests for COVID-19 at that time.” However, I do not think it is acceptable to change the original question.

Response:

Thank you for your comment. 

We totally agree with your comment. We have never changed the expression and meaning of the original questions. We are afraid that it is misunderstanding due to our inaccurate and confusing reply to the inquiry from the editor, that is “We have revised the questionnaire as part of the second review process.” Obviously, it was not correct. The truth is that our translation from Japanese statement in the original questionnaire to English explanation in our manuscript, replies to the second reviewer, and questionnaire in the supplementary data were wrong. Please understand what we revised was not the original questionnaire but only an English translation.

Specifically, we found a careless mistake of the translation in the question No. F15-4 (the original question in Japanese language “検査がすぐに受けられないこと”): The first author translated it as “Not being able to take the PCR test immediately”. However, during review process, the co-authors noticed that the translation was incorrect as the word “the PCR test” is not accurate and inconsistent with the original question in Japanese. Thus, we revised the translation as “Inability to receive COVID-19 tests immediately” in the manuscript and replies to the second reviewer as well as the English-translated questionnaire in the supplementary data. 

We sincerely apologize for the confusion we have caused. 

2. Supplemental data for all the response data were not included in this revision, while PLOS ONE asked authors to upload their data somewhere. I found the data in revision R1. I did not know why authors deleted them.

Response:

Thank you very much for your comment. We apologize for not knowing that we had to submit the data set in every revision. As you mentioned that we submitted the data set for the first time, we thought that it was not necessary to submit it again. However, we would like to resubmit the data set including the revised translation of the questionnaire with this revised manuscript. 

Minor points:

3. P.17 L.229 Regular employees among females (AOR 1.125 …)

While in Table 3 (P.20), it is “Non regular” employee (AOR “1.215”). Please confirm both the label and the number, and fix either one.

Response:

Thank you for pointing out the mistake. The number in Table 3 is correct. The number of the manuscript (AOR 1.125) has been revised.

4. P.17 L.230-232 there are typos. Shouldn't all “≥ 6 million” be “< 6 million?” (according to Table 3.)

Response:

Thank you for pointing out the mistake. Three parts of the manuscript have been corrected.

5. P.20 Table 3 Some names of rows were difficult to be understood and different from the questionnaire, e.g., “What I may be infected with the virus,” “What may affect my family.”

Response:

Thank you for your comment. 

We apologize for the incomprehensibleness and the differences between the questionnaire and the names of the rows in Table 3. We renamed the names of the rows in Table 3 and also Table 1, so that the names become consistent with the translated questionnaire. We also revised several wordings so that they become consistent among the manuscript, tables, and the translated questionnaire. 

6. P.22 L.326 “Three previous studies in China and Turkey also reported that Chinese female …”

The study from Turkey (ref. 6) should not report Chinese results. Therefore, please remove “Chinese” here.

Response:

Thank you for your suggestion. We removed the word “Chinese”. 

7. P. 22-23 “Previous research concluded that females were at a higher risk of mental health problems than men (31).”

Ref. 31 is a paper on post-injury mental health problems. It seems to be a particular case and a little far from the current results. Could the authors revise this part? 

Also, in the following sentence, “these three previous studies” is written. It is better to remove “three,” if the authors mention both Ref 15 and 31, together with Ref 3, 6, and 30.

Response:

Thank you for your suggestion. We deleted the line mentioned above and the word “three”.

8. P. 28 L.430 “using a PC device” via the internet.

To my knowledge, such kind of survey can also be answered by smartphones. If so, please remove using a PC device or add some other means to make it clear.

Response:

Thank you for your suggestion. We removed the word “using a PC device” and added a sentence “a device which can connect to the Internet”.

---

## [Decision Letter · Decision Letter 3]

10 Feb 2021

PONE-D-20-21244R3

Impacts of anxiety and socioeconomic factors on mental health in the early phases of the COVID-19 pandemic in the general population in Japan: A web-based survey

PLOS ONE

Dear Dr. Nagasu,

Thank you for submitting your manuscript to PLOS ONE. After careful consideration, we feel that it has merit but does not fully meet PLOS ONE’s publication criteria as it currently stands. Therefore, we invite you to submit a revised version of the manuscript that addresses the points raised during the review process.

The reviewer addressed two minor concerns about your manuscript.  Please send me the revised manuscript ASAP. I will make the final decision (accept) without peer review.

We look forward to receiving your revised manuscript.

Kind regards,

Kenji Hashimoto, PhD

Academic Editor

PLOS ONE

Reviewers' comments:

Reviewer's Responses to Questions

**Comments to the Author**

1. If the authors have adequately addressed your comments raised in a previous round of review and you feel that this manuscript is now acceptable for publication, you may indicate that here to bypass the “Comments to the Author” section, enter your conflict of interest statement in the “Confidential to Editor” section, and submit your "Accept" recommendation.

Reviewer #3: All comments have been addressed

2. Is the manuscript technically sound, and do the data support the conclusions?

Reviewer #3: Yes

3. Has the statistical analysis been performed appropriately and rigorously? 

Reviewer #3: Yes

4. Have the authors made all data underlying the findings in their manuscript fully available?

Reviewer #3: Yes

5. Is the manuscript presented in an intelligible fashion and written in standard English?

Reviewer #3: Yes

6. Review Comments to the Author

Reviewer #3: Dear editor and authors,

Thank you for giving me a chance to reconfirm this manuscript: “Impacts of anxiety and socioeconomic factors on mental health in the early phases of the COVID-19 pandemic in the general population in Japan: A web-based survey.” It is very interesting and meaningful to understand the citizens’ mental health in the early stage of the COVID-19 pandemic. The data are sampled per the Japanese population and have a sufficient number of respondents to give the results and conclusion.

I have reconfirmed the revision of this manuscript, and most of the questions have been answered. Although two minor points are listed below, I think they can be fixed during the publication steps. Therefore, now, I advise the publication of this manuscript in PLOS ONE.

Minor points:

1. P. 17 L. 229 “such as regular employees among females…”

Shouldn’t it be non-regular employees? According to Table 3, regular employees are the reference. The odds ratio is non-regular employees/regular employees.

2. P. 8 L.136 “Have you been worried the following items …”

“about” should be added after “worried” for English grammar.

7. PLOS authors have the option to publish the peer review history of their article (what does this mean?). If published, this will include your full peer review and any attached files.

Reviewer #3: **Yes: **Yu-Shi Tian

---

## [Author Response · Author response to Decision Letter 3]

11 Feb 2021

Response to reviewers:

PLOS ONE

PONE-D-20-21244

Impacts of anxiety and socioeconomic factors on mental health in the early phases of the COVID-19 pandemic in the general population in Japan: a Web-based survey

6. Review Comments to the Author

Reviewer #3: Dear editor and authors,

Thank you for giving me a chance to reconfirm this manuscript: “Impacts of anxiety and socioeconomic factors on mental health in the early phases of the COVID-19 pandemic in the general population in Japan: A web-based survey.” It is very interesting and meaningful to understand the citizens’ mental health in the early stage of the COVID-19 pandemic. The data are sampled per the Japanese population and have a sufficient number of respondents to give the results and conclusion.

I have reconfirmed the revision of this manuscript, and most of the questions have been answered. Although two minor points are listed below, I think they can be fixed during the publication steps. Therefore, now, I advise the publication of this manuscript in PLOS ONE.

Response:

We appreciate the reviewer to review our manuscript and give us productive comments. For the minor points suggested below, we modified the manuscript according to the reviewer’s comments. 

Minor points:

1. P. 17 L. 229 “such as regular employees among females…”

Shouldn’t it be non-regular employees? According to Table 3, regular employees are the reference. The odds ratio is non-regular employees/regular employees.

Response:

We added the word “non-” and the sentence is as follows: 

“as non-regular employees among females (AOR1.215 [95% CI: 1.025–1.440])” 

2. P. 8 L.136 “Have you been worried the following items …”

“about” should be added after “worried” for English grammar.

Response:

Thank you for your suggestion. We added the word “about” and the sentence is as follows: 

“Have you been worried about the following items after the outbreak of the new coronavirus infection?”

---

## [Editor Report · Decision Letter 4]

12 Feb 2021

Impacts of anxiety and socioeconomic factors on mental health in the early phases of the COVID-19 pandemic in the general population in Japan: A web-based survey

PONE-D-20-21244R4

Dear Dr. Nagasu,

We’re pleased to inform you that your manuscript has been judged scientifically suitable for publication and will be formally accepted for publication once it meets all outstanding technical requirements.

Kind regards,

Kenji Hashimoto, PhD

Section Editor

PLOS ONE
---

## [Editor Report · Acceptance letter]

17 Feb 2021

PONE-D-20-21244R4 

Impacts of anxiety and socioeconomic factors on mental health in the early phases of the COVID-19 pandemic in the general population in Japan: A web-based survey 

Dear Dr. Nagasu:

I'm pleased to inform you that your manuscript has been deemed suitable for publication in PLOS ONE. Congratulations! Your manuscript is now with our production department. 

Kind regards, 

on behalf of

Prof. Kenji Hashimoto 

Section Editor

PLOS ONE